# Groundwater level observations in 250,000 coastal US wells reveal scope of potential seawater intrusion

Scott Jasechko [1✉], Debra Perrone [2], Hansjörg Seybold[3], Ying Fan [4] & James W. Kirchner [3,5,6]

Seawater intrusion into coastal aquifers can increase groundwater salinity beyond potable levels, endangering access to freshwater for millions of people. Seawater intrusion is particularly likely where water tables lie below sea level, but can also arise from groundwater pumping in some coastal aquifers with water tables above sea level. Nevertheless, no nationwide, observation-based assessment of the scope of potential seawater intrusion exists. Here we compile and analyze ~250,000 coastal groundwater-level observations made since the year 2000 in the contiguous United States. We show that the majority of observed groundwater levels lie below sea level along more than 15% of the contiguous coastline. We conclude that landward hydraulic gradients characterize a substantial fraction of the East Coast (>18%) and Gulf Coast (>17%), and also parts of the West Coast where groundwater pumping is high. Sea level rise, coastal land subsidence, and increasing water demands will exacerbate the threat of seawater intrusion.

[1] Bren School of Environmental Science and Management, University of California at Santa Barbara, Santa Barbara, CA 93106, USA. [2] Environmental Studies Program, University of California at Santa Barbara, Santa Barbara, CA 93106, USA. [3] Department of Environmental System Sciences, ETH Zürich, Universitätstrasse 16, Zürich CH-8092, Switzerland. [4] Department of Earth and Planetary Sciences, Rutgers University, New Brunswick, NJ 08854, USA. [5] Swiss Federal Research Institute WSL, Birmensdorf CH-8903, Switzerland. [6] Department of Earth and Planetary Science, University of California, Berkeley, CA 94720, USA. ✉email: jasechko@ucsb.edu

Seawater intrusion threatens freshwater resources by rendering coastal groundwaters too saline for drinking or irrigation[1,2]. Over ~100 million Americans and thousands of farms in coastal counties depend fully or partly on groundwater[3]. Well water can be impacted by even small amounts of seawater intrusion: groundwater containing more than 2–3% seawater is considered non-potable. Aquifer salinization by seawater is almost irreversible on human timescales, because the intruded seawater occupies small pore spaces that can require decades or centuries to be flushed[4,5]. Consequently, it is important to identify aquifers that are susceptible to seawater intrusion to inform management actions.

Seawater intrusion can occur naturally or be induced as groundwater is pumped from wells. Even under static, pre-development conditions one would expect seawater to exist at depth beneath low-lying coastal lands, because seawater is denser than freshwater and because tidal variations disperse the fresh–saline interface in coastal aquifers[1,2]. Climate and land-use changes can reduce recharge, lower groundwater levels, and induce seawater intrusion. Overpumping can lower groundwater levels below sea level, leading to hydraulic gradients that slope downward toward the land (herein *landward hydraulic gradients*). A landward hydraulic gradient implies that seawater intrusion could occur if the coastal aquifer is well connected to the sea. Identifying locations with landward hydraulic gradients can reveal which aquifers are susceptible to seawater intrusion, because hydraulic gradients drive groundwater flow and influence the depth at which aquifers transition from fresh to brackish water[6–8]. Seawater intrusion can occur even before landward hydraulic gradients form, because seawater's higher density can cause it to move landward even if coastal water tables are above sea level.

To date, our understanding of seawater intrusion is founded on either (i) local-scale studies (Fig. 1; ref. [9]) or (ii) hydrologic models[10,11].

(i) Local-scale studies provide the foundation of current conceptual models of coastal aquifers and seawater. The 108 local-scale studies compiled in Supplementary Table 5 demonstrate important features that motivate our study: (a) areas with high topographic relief and high recharge rates can maintain water tables above sea level, inducing groundwater discharge into the sea and limiting seawater intrusion (e.g., Puget Sound[12–14]; Fig. 1a); (b) extensive groundwater pumping can lead to seawater intrusion (e.g., Salinas Valley[15–17]; Fig. 1b), making seawater intrusion relatively common in low-lying alluvial valleys on the West Coast (e.g., Los Angeles[18,19]; Fig. 1c) and along the Gulf[20–23] and East[24–40] Coasts where extensive coastal plain aquifers are tapped for irrigation and public water supplies (Fig. 1d–g); and (c) seawater intrusion occurs in both shallower unconfined and deeper confined aquifers, and, in many places, the deeper aquifers are more vulnerable due to their isolation from local recharge and low storativity, making hydraulic heads sensitive to pumping. These insights provide a conceptual framework that demonstrates how seawater intrusion depends on climate, terrain and groundwater use, and imply that seawater intrusion assessments should include both shallow and deep aquifers (Fig. 1). A common theme among the local-scale studies is that the rates and spatial patterns of intruding seawater depend not only on hydraulic gradients but also on the heterogeneity and architecture of the coastal aquifer system (e.g., connectivity of sedimentary layers to the sea, presence of *windows* (i.e., gaps) in aquitards).

(ii) Because local-scale studies do not cover the whole coastline continuously (Fig. 1) the potential for seawater intrusion has not been systematically assessed along the nation's coastline. In an attempt to fill this knowledge gap, hydrologic models have been used to estimate seawater intrusion along contiguous US coastlines[10,11]. Sawyer et al.[11] simulate the catchment water balance to depths of ~30 m below the land surface and compare their model results against local-scale studies. In some areas, seawater intrusion is largely constrained to the uppermost ~30 m, making this a suitable depth to evaluate (e.g., Miami[39,40]; Fig. 1e). However, local-scale studies (Fig. 1) demonstrate the three-dimensional nature of seawater intrusion in layered aquifer systems, where hydraulic heads and the advance of seawater differ between shallower versus deeper aquifers, just as they may differ laterally at different locations along the coastline (e.g., the distance seawater has intruded inland differs between the *180-foot Aquifer* and the *400-foot Aquifer* in California's Salinas Valley[15–17]; Fig. 1b). These studies demonstrate seawater intrusion into confined aquifers lying deeper than the ~30 m depth simulated by Sawyer et al.[11] (e.g., the *Atlantic City 800-foot Sand* in Cape May[35]; Fig. 1g). Because current model-based assessments of submarine groundwater discharge and seawater intrusion focus mostly on shallow unconfined aquifers, we lack a continental-scale assessment of the potential for seawater intrusion at depths deeper than ~30 m. Further, some continental-scale coastal hydrogeologic models embed assumptions that cannot be straightforwardly evaluated, such as an assumed relationship between population density and groundwater pumping.

Here, we assess the potential for seawater intrusion at continental-scale, by collating and analyzing 250,000 groundwater well water observations made since the year 2000 within 10 km of the coast (Supplementary Note 1). The 1st–99th percentile range of well depths is 3–205 m, meaning our dataset samples both unconfined and confined aquifers. The unprecedented spatial resolution of our dataset, both laterally and vertically, allows for observation-based estimates of landward hydraulic gradients along the coastline of the lower 48 states of the US. Our results use observational data to generate locally relevant predictions using a consistent methodology in areas that have not been the focus of local-scale studies (for a detailed comparison with previous work see Supplementary Note 8). Most of our measurements are *static water levels* measured since 2000 that derive from drilling reports (i.e., measurements made before pumping); we show that these measurements agree with nearby monitoring well readings, implying that they provide reliable data on actual aquifer conditions (Supplementary Note 3). The density of our well observations and the wide range of aquifer depths make our approach uniquely suited to address the knowledge gaps outlined above. Our objective is to identify coastal areas where most well water elevations lie below sea level, rendering these areas potentially vulnerable to seawater intrusion.

## Results and discussion

**Observations reveal scope of potential seawater intrusion.** Areas where most well water elevations lie below sea level—indicative of landward-directed hydraulic gradients—are found on all three coastlines (Fig. 2). Reported well water depths were converted to elevation above sea level using a ~10 m-gridded digital elevation dataset (available from ned.usgs.gov). For monitoring wells reporting more than one water level measurement, we include only the median of all measurements made from 2000–present in all results to follow. More than half of all recorded well water elevations that are below sea level in our compiled dataset are found within 8 km of the coast (Fig. 3a, b), and 70% are within ~20 km of the coast (Fig. 3b). Well water elevations that are below sea level and located more than 20 km inland are mainly found along the Gulf Coast and in areas where the land surface itself lies below sea level. Landward hydraulic gradients are more common along the Gulf (South) and Atlantic (East) Coasts relative to the Pacific (West) Coast (Fig. 4a–c), as

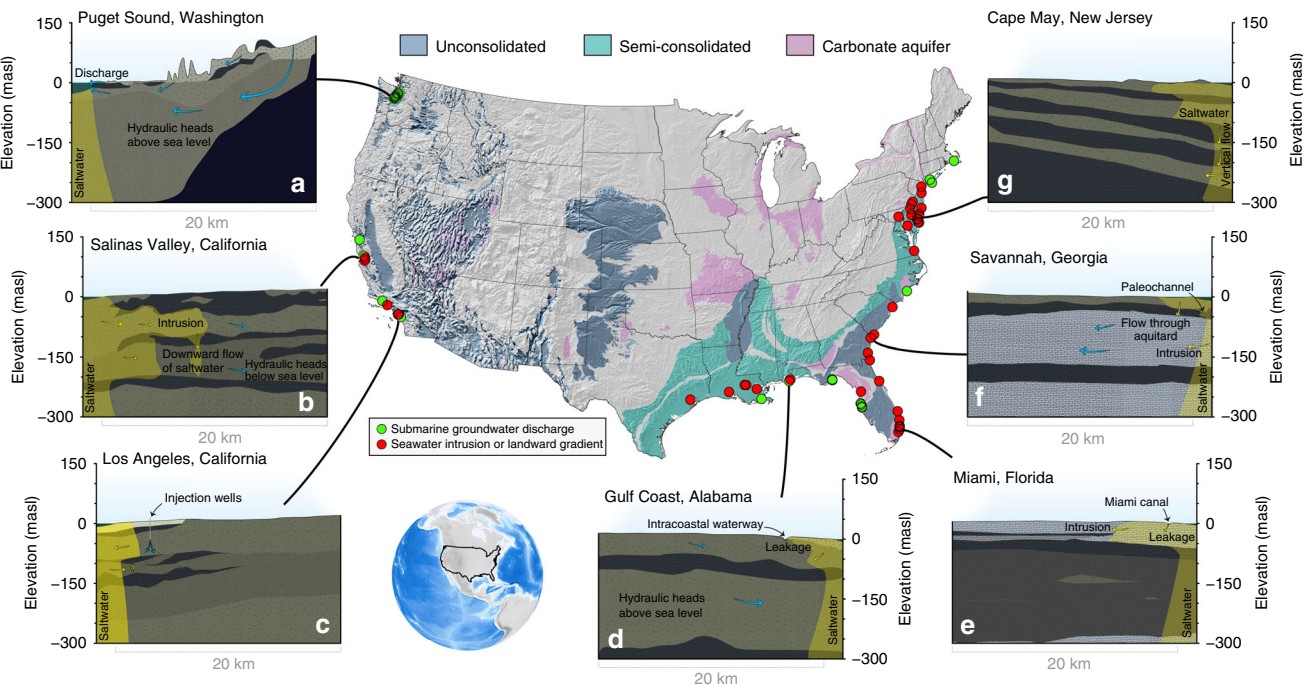

**Fig. 1 Local-scale studies of submarine groundwater discharges and seawater intrusion.** These local-scale studies use a variety of approaches, including hydrological models, seepage meters, and solute concentration measurements (see review by Santos et al. [65]). We show locations where local-scale studies have demonstrated or simulated the occurrence of submarine groundwater discharge (blue points), or a landward gradient or seawater intrusion (red points). For primary results and discussion of important local considerations see refs. [66–68]. For reviews of studies of submarine groundwater discharge research and seawater intrusion prior to the year 2000 see refs. [9,69]. For primary references see Supplementary Table 5), which reviews 108 studies. We present a series of conceptual models based on previous studies as cross sections. **a** In Puget Sound[12–14] (Washington state), high relief, high recharge rates and modest groundwater use create seaward hydraulic gradients and concomitant submarine groundwater discharge. **b** In the Salinas Valley[15–17] (California), decades of extensive groundwater pumping and limited local recharge (due to a shallow aquitard) have increased landward hydraulic gradients and induced seawater intrusion. Seawater has moved several kilometers inland in both the shallow (180-foot) aquifer and the deeper (400-foot) aquifer. **c** In the West Coast Basin[18,19] (Los Angeles, California), high groundwater withdrawals have created landward hydraulic gradients and seawater intrusion, but the installation and operation of injection wells have increased hydraulic heads in the aquifer close to the coast. **d** In the Gulf Coast of Alabama[20], hydraulic heads lie just above sea level yet some intrusion has occurred. **e** Near Miami[39,40] (Florida), a shallow carbonate aquifer (Biscayne) has experienced seawater intrusion, likely exacerbated by the construction of leaky canals. **f** In Savannah[27] (Georgia), groundwater withdrawals have drawn down a piezometric surface that once lay above sea level to now lie below sea level, creating a landward hydraulic gradient. **g** In Cape May[35] (New Jersey), landward gradients are clearly reflected in piezometric data for confined aquifers (e.g., the Cohansey and Atlantic City 800-foot sand), highlighting the potential vulnerability of confined aquifers to seawater intrusion. Aquifer outlines are from water.usgs.gov/ogw/aquifer/map.html.

discussed below. We report our results as percentages of wells exhibiting landward gradients within segments 20 km long (along the shore) and 10 km wide (extending inland). These segments allow us to report regional patterns of landward hydraulic gradients based on point observations (Table 1). Segments are characterized as exhibiting landward hydraulic gradients if more than half of their well water levels are lower than sea level. Where well water elevations are equal to or slightly above the sea level, hydraulic gradients may be seaward, or may still be landward because of the higher density of seawater (i.e., the equivalent hydraulic head resulting from the Ghijben–Herzberg principle). Furthermore, a small drop in water table elevations—which may still lie above sea level—can lead to substantial landward movements of seawater where topographic relief is minimal. As a result, our approach—detecting landward hydraulic gradients and interpreting these as precursors to seawater intrusion—is conservative. Seaward hydraulic gradients can occur even where most well water elevations lie below sea level, if a hydraulic ridge exists (e.g., if well water elevations lie below sea level 5–10 km inland, but are above sea level within 5 km of the coast). Therefore, we evaluate the sensitivity of our results to the distance from the coast (i.e., measurements within 10 km of the coast versus only those within 1 km of the coast; Supplementary Note 4), and we

find that our main conclusions remain valid within this range of choices.

Because some deeper wells tap confined aquifers, adjacent shallow and deep wells can yield different water elevations. We cannot identify depths at which aquifer systems transition from unconfined (shallow wells) to confined (deeper wells) conditions, because three-dimensional continent-wide geologic datasets are currently unavailable. Instead, we present a sensitivity analysis, based on re-running our analyses using only wells shallower than six different threshold depths (e.g., results based only on wells shallower than 100 m versus results based only on wells deeper than 100 m; Supplementary Note 4). In general, we find that well water elevations that lie below sea level are more common in deeper wells, but that the general spatial patterns of well water elevations in coastal aquifers (i.e., Figs. 2 and 3) hold across a wide range of threshold well depths (Supplementary Figs. 9–14 and Supplementary Table 2).

**West Coast**. About 15% of West Coast well water elevation observations within 10 km of the coast lie below sea level (Table 1; Fig. 2). Well water elevations that are below sea level are concentrated in densely populated or heavily irrigated parts of the coast, such as central and southern California (e.g., Monterey,

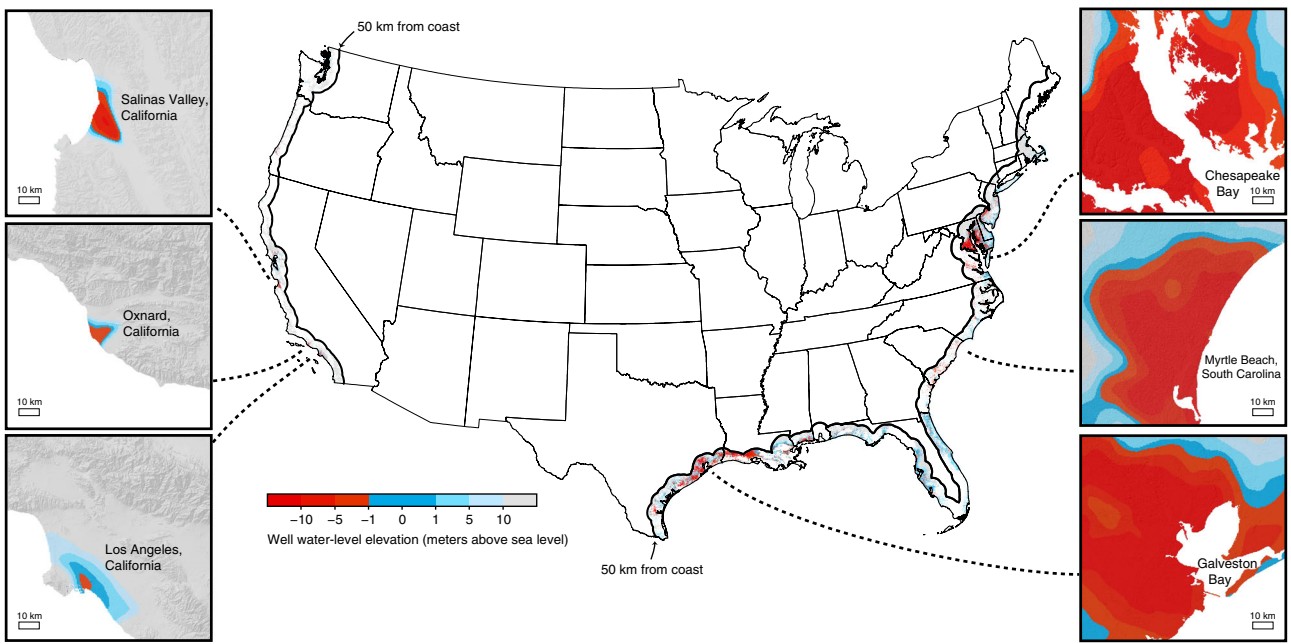

**Fig. 2 Well water elevations across the contiguous United States.** Each point on the center map represents a unique well, and its color corresponds to the well water elevation measured more recently than January 1, 2000. Red points correspond to well water elevations that are more than one meter below sea level. For monitoring wells reporting more than one well water elevation measurement, we show the median of all measurements made after the year 2000. We only present data for wells located within 50 km of the coast. Well water elevations below mean sea level (i.e., red points) are concentrated in densely populated parts of California's coast (Los Angeles), much of the west-central part of the Gulf (south) coast, and portions of the Atlantic Coast (e.g., western Delmarva Peninsula). Six 100 km by 100 km maps are shown for specific areas of each coastline (red-blue shades are interpolations of well water elevations; interpolated surfaces follow the same color scale as point data in continent-wide map).

Oxnard, Los Angeles). Among all studied 20-km coastline segments with sufficient data (see "Methods" section), 4.2% have over half of their well water elevation measurements below sea level (Fig. 4a). Landward hydraulic gradients characterize at least 2.4% of the US West Coast (Table 1). Because well locations are particularly uncertain in California, we conducted sensitivity analyses for the West Coast and found that at least half of well water elevations could lie below sea level for as many as 6.9% of all studied West Coast segments (Table 1; see Supplementary Note 6 and Supplementary Table 4).

**Gulf Coast**. Nearly one-quarter (22.6%) of Gulf Coast well levels within 10 km of the coast lie below sea level (Table 1; Fig. 2). At least half of all well water elevations are below sea level in 40.1% of studied coastline segments (Fig. 4b). At least half of all measured well water elevations are below sea level near Houston (Texas), New Orleans (Louisiana), Gulfport (Mississippi), Panama City, St. Petersburg, and Venice (northwest and west Florida). Seaward hydraulic gradients are common to most of northwestern Florida, whereas landward hydraulic gradients characterize much of the western portion of the Gulf Coast. Our analysis reveals substantial variability in hydraulic gradients from west to east along the Gulf Coast (Fig. 4b). We conclude that landward hydraulic gradients characterize at least 17.3% of the US Gulf Coast (Table 1).

**East Coast**. About one-third (34.7%) of East Coast well water elevations measured within 10 km of the coast are below sea level (Fig. 2). At least half of all measured well water elevations are below sea level for 38.8% of studied coastline segments (see "Methods" section; Fig. 4c). Areas where well water elevations are frequently below sea level include Miami (Florida), Savannah (Georgia), Myrtle Beach (South Carolina), Virginia Beach (Virginia), west-facing and east-facing shorelines of Chesapeake Bay

(Maryland), and Somers Point (New Jersey; Fig. 4c). Conversely, most well water elevations are above sea level along north Chesapeake Bay shorelines and along most of the coastlines of New Jersey, New York, Massachusetts and New Hampshire (Fig. 4c). Landward hydraulic gradients characterize at least 18.4% of the US East Coast (Table 1).

**Seawater intrusion threatens American aquifers**. Well water elevations suggest that many coastal aquifers are threatened by seawater intrusion arising from landward hydraulic gradients (Fig. 2). We find that landward hydraulic gradients are common along at least 15% of the contiguous United States coastline, are more common along the Gulf and East Coasts than along the West Coast (Fig. 4), and exhibit high spatial variability along all coasts. Hydraulic gradients conducive to seawater intrusion are common to several major US population centers such as Houston and Los Angeles (Figs. 2 and 4). We emphasize that well waters can become salinized via seawater intrusion long before landward hydraulic gradients emerge, wherever pumping has directly drawn saline water upward from deeper parts of a coastal aquifer.

**Landward hydraulic gradients threaten coastal groundwater use**. The main finding of our research is that US coastal aquifers —relied on by many large population centers and agricultural areas—are extensively threatened by seawater intrusion. Landward hydraulic gradients exist along at least 15% of the US coastline, and many of these areas encompass urban centers where fresh groundwater is critical for household uses (Figs. 2–4). Such landward gradients imply that the potential for seawater intrusion exists, but the gradients by themselves do not demonstrate that seawater intrusion is occurring. Many factors beyond just hydraulic gradients control the likelihood of seawater intrusion. These factors include aquifer types (unconfined, partially

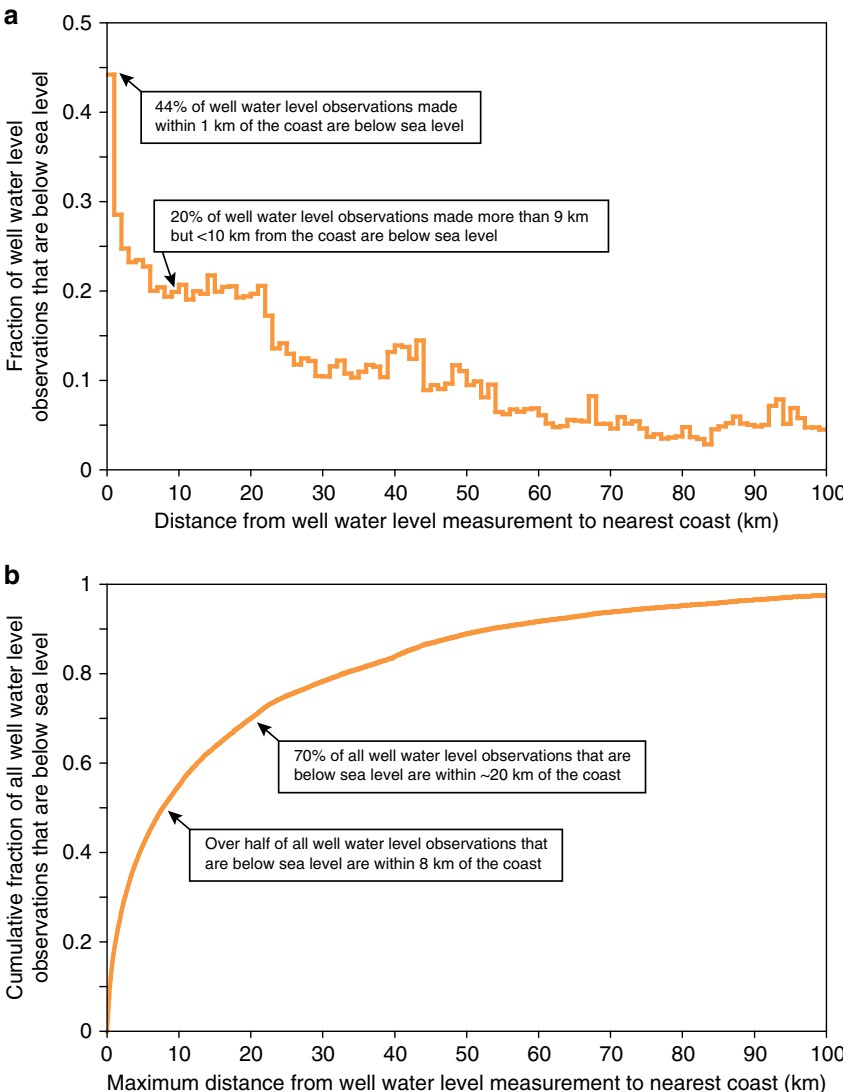

**Fig. 3 Fraction of well water elevations that are below sea level, as a function of distance from the coastline. a** Fraction of well water elevations below sea level, as a proportion of all recorded well levels at varying distances to the nearest coast. **b** Cumulative fraction of all well water elevations that are below sea level. More than half of all wells with water elevations that lie below sea level are within 10 km of the coast. Each well is counted once; for wells with multiple measurements postdating 2000, we calculated median well water elevations for 2000–present and incorporate only this median value in our calculations to produce the above figures.

confined, confined), hydraulic conductivities, aquifer and aquitard thicknesses, and recharge rates (see refs. [41,42]).

Seawater intrusion can occur even where groundwater levels lie above sea level, as depths to coastal freshwater–saltwater interfaces can be <~100 m, where well water elevations are 1–3 m above sea level and groundwater pumping induces an upwelling of saltwater from these deeper depths (e.g., see ref. [43]). Further, heterogeneity in aquifer flow paths and connectivity to surface processes mean that hydraulic heads may switch from below to above sea level over short lateral distances along coastlines (~kilometers); thus, even in sections of coastline where the great majority of well water elevations lie below sea level, not all wells are necessarily vulnerable to seawater intrusion. Influences from relict climate conditions and past local sea level rise may mean that pre-development fresh–saline groundwater interfaces had not yet reached equilibrium[44], implying seawater intrusion was occurring in some places even before the installation of the first US water well in the early 1800s[44].

We highlight that our analysis assumes that most well water elevation measurements are made in wells filled with freshwater. This assumption is relevant because differences in fluid density can influence hydraulic heads (see ref. [45]). Most of our well water elevation measurements originate from groundwater well completion reports, many of which are linked to beneficial uses of fresh groundwater; therefore, the recorded well levels likely reflect conditions in aquifers that bore fresh water at the time of well construction. Our results indicate where seawater intrusion may have already occurred, or may occur in the future because landward hydraulic gradients exist.

We also stress that seawater intrusion is just one of several processes that lead to groundwater salinization. Others include dissolution of evaporite minerals[46] (e.g., halite, gypsum), mixing with naturally occurring brines[47], infiltration of seawater reaching the land surface by storm surges or tsunamis[48–51], mixing with seawaters emplaced during marine high-stands[52] (i.e., when local sea levels were higher than present), infiltration of salts derived from dry and wet deposition of airborne particles[53], percolation

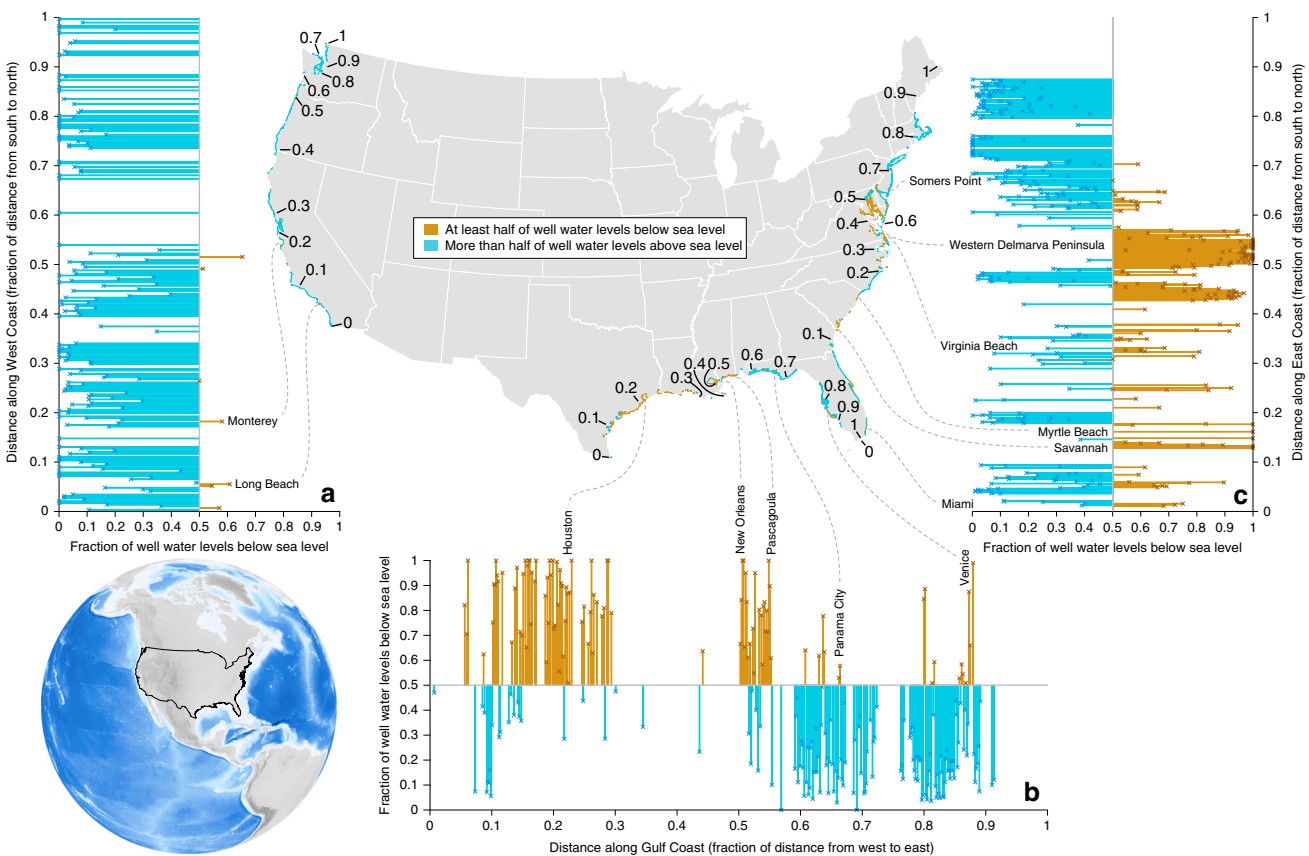

**Fig. 4 The fraction of well water elevations that lie below sea level along contiguous US coasts.** The center map displays distances along each coast. Colors along the coastlines shown on the center map represent 20-km segments of coast, reaching 10 km inland; orange and blue segments represent those where more than half of the measured well levels are below and above sea level, respectively. The three bar plots present the fraction of well water elevations that are below sea level; each bar represents one 20-km coastline segment, reaching 10 km inland. The numbers on the map indicate the fractional distance along the coastline from south to north (panels **a** and **c**) or west to east (panel **b**), which can be followed in the corresponding bar plots. Selected coastal cities and regions where at least half the nearby well water elevations are below sea level are individually labeled. The inset map shows the location of the contiguous US (black outline). Because well water elevations that lie below sea level are more common closer to the coast (Fig. 3), we also reproduced Fig. 4 using three additional distance thresholds to assess their effects on our findings (see Supplementary Note 4, Supplementary Table 1, and Supplementary Figs. 6–8). Further, we also present alternate versions of Fig. 4 that separate shallower from deeper wells to test the sensitivity of our results to well depth (Supplementary Figs. 9–14).

| Table 1 Well water elevations along west, gulf and east coasts. | | | | | | |
|---|---|---|---|---|---|---|
| Coastline | Wells with water elevation measurement(s) within 10 km of coast | Wells within 10 km of coast with water elevation below sea level | Percent of wells within 10 km of coast with water elevation below sea level | Total number of segments (20-km-long) along coast | Number of segments (20-km-long) where more than half of water elevations are below sea level | Percent of segments (20-km-long) where more than half of water elevations are below sea level |
| Lower 48 states | 256,750 | 69,153 | 26.9 | 1765 | 272 | 15.4 |
| West coast | 14,196 | 2182 | 15.4 | 291 | 7 | 2.4[a] |
| Gulf coast | 142,141 | 32,125 | 22.6 | 589 | 102 | 17.3 |
| East coast | 100,413 | 34,846 | 34.7 | 885 | 163 | 18.4 |
| [a]May be as high as 6.9%—see Supplementary Note 6 and Supplementary Table 4. | | | | | | |

beneath tidal marshes[40], and groundwater recharge impacted by surface activities (e.g., urban road salting, agricultural practices; refs. [5,54]). Pumpage can also induce salinization via upconing, where deep saline waters upwell as a result of pumping from shallower aquifers (see research[26] on the upwelling of saline waters from the Fernandina Permeable Zone into the Floridan Aquifer near Brunswick, Georgia).

**Strategies to manage and monitor seawater intrusion.** Approaches to managing and observing seawater intrusion fall

into three broad categories: (i) regulation, (ii) monitoring, and (iii) engineering.

(i) Overpumping of aquifers is a leading driver of seawater intrusion in many areas. Limiting groundwater pumping via regulatory mechanisms can help groundwater levels stabilize or rebound where they have dropped below sea level, potentially slowing or stopping seawater intrusion. For example, in Monterey County, local agencies have the power to regulate groundwater extraction to prevent seawater intrusion (Monterey County Water Resources Agency Act § 52-22; ref. [55]). In California more broadly, landward gradients are concentrated in densely populated and irrigated areas. Under California's 2014 Sustainable Groundwater Management Act, Groundwater Sustainability Agencies across the state must consider seawater intrusion as they develop their plans for managing and using groundwater (e.g., ref. [15]). Nevertheless, such regulatory mechanisms may be most useful for preventative management: once an aquifer is contaminated it could take decades to flush out salts or reverse intrusion fully, due to the long timespans required to completely flush intruded seawater from an aquifer. In these cases, engineered controls may be required to ameliorate salinization (see (iii) below).

(ii) Groundwater level and groundwater quality monitoring is important in many coastal areas. Nevertheless, few states mandate metering, monitoring, *and* reporting information associated with groundwater use[56]. Establishing, continuing, or augmenting monitoring activities can help to quantify the extent and pace of seawater intrusion or provide vital information about impending intrusion. Monitoring also can be used alongside engineering solutions to assess progress (e.g., injection wells to keep seawater at bay—see West Coast Basin[19], Fig. 1b). Although our analysis represents the most extensive observation-based assessment of coastal hydraulic gradients to date, further monitoring is warranted in many areas where data remain scarce. We evaluated well water elevation variations in coastal monitoring wells across the US, and show that these monitoring well records are valuable for identifying coastal aquifers where (i) landward hydraulic gradients currently exist, but well water elevations are increasing over time (e.g., northwestern shores of Galveston Bay; landward gradient shown in Fig. 2; shallowing well water elevation trend shown in Supplementary Fig. 20); (ii) landward gradients currently exist and well water elevations are decreasing with time (e.g., western shores of Chesapeake Bay; landward gradient shown in Fig. 2; deepening well water elevation trend shown in Supplementary Fig. 20), and (iii) most well water elevations are above sea level, but well water elevations are declining, implying a landward gradient may arise if levels continue to decline (e.g., Santa Barbara, California; seaward gradient shown in Fig. 2; deepening well water elevation trend shown in Supplementary Fig. 20).

(iii) Controlling hydraulic gradients via engineering can slow seawater intrusion or help reverse landward hydraulic gradients. One engineering approach involves injecting freshwater into wells to create localized hydraulic barriers, reversing landward hydraulic gradients back to natural seaward hydraulic gradients. For example, in the Los Angeles Basin, three lines of injection wells have been constructed, successfully slowing seawater intrusion in some portions of the aquifer system[18]. Actively inducing recharge in carefully selected areas via spreading basins can also help create hydraulic barriers that slow seawater intrusion. Desalination and water recycling technologies can reduce water demands, and therefore help slow seawater intrusion by reducing demands for groundwater. Extracting and desalinating seawater for managed aquifer recharge has also been proposed to limit seawater intrusion[57]. In parts of California, excess surface water, stormwater, and wastewater are used for managed aquifer recharge projects with the intention of creating a barrier to seawater intrusion[58]. While these approaches may prove suitable in densely populated areas with capital to invest in infrastructure, they are unlikely to be feasible solutions for the whole ~5000 km of US coastline affected by landward hydraulic gradients (Fig. 4).

**Potential for seawater intrusion along much of the coast.** Expanding access to secure sustainable fresh water supplies remains a great challenge of the 21st century. Groundwater resources are often viewed as being more resilient to climate variability than surface water supplies[59,60], and are already used widely. We show that well water measurements reported in well completion reports frequently provide high-density and high-quality hydraulic gradient information (Supplementary Notes 2 and 3). Analyzing coastal well water elevation data can help to identify coastal aquifers that are vulnerable to seawater intrusion, and also to improve assessments of coastal groundwater discharges and concomitant marine solute influxes. For this study, we define aquifers as particularly vulnerable to seawater intrusion if most coastal well water elevations lie below sea level. We re-emphasize that seawater intrusion can occur even where most well water elevations lie above sea level, meaning our main finding—that 15% of US coastlines are characterized by landward hydraulic gradients—probably underestimates the prevalence of vulnerability to seawater intrusion.

Our analyses of landward hydraulic gradients highlight that larger proportions of the United States' tectonically passive continental margins—i.e., the East and Gulf Coasts—are affected relative to the West Coast. Even where most or all well water elevations lie below sea level, it may take decades for seawater to move kilometers inland, as the hydraulic conductivity of aquifers limits groundwater flow speeds. Thus, if vulnerable aquifers can be identified in time, the worst impacts of seawater intrusion can potentially be avoided. To the best of our knowledge, this study is the first to analyze densely distributed continental-scale well water elevation observations, allowing us to develop a measurement-driven national assessment of vulnerability to seawater intrusion. Active monitoring and new regulatory or engineering interventions in vulnerable locations can potentially slow or stop seawater intrusion and protect coastal groundwater quality.

## Methods

**Data synthesis**. We compiled well water measurements from two types of data sources: (i) well water measurements recorded in well completion reports[61], digitized and provided by state or regional water agencies (herein *constructed wells*; Supplementary Note 1), and (ii) monitoring well records maintained by the United States Geological Survey (USGS) or the Groundwater Ambient Monitoring & Assessment Program (GAMA) of California's State Water Resources Control Board (herein *monitoring wells*).

Results presented in the main text focus mostly on well water elevations measured near the coast. To evaluate the quality of water levels reported in well completion reports (see the next subsection and Supplementary Note 3), we also compiled well water elevation data for the non-coastal contiguous United States, Alaska, and Hawaii.

**Quality assurance of water levels in well completion reports**. Before analyzing well water level measurements in well completion reports, we assessed how they compared to well water level measurements made under similar hydrogeologic conditions in nearby monitoring wells (Supplementary Note 3). Specifically, we compared water level measurements reported in well completion reports versus those reported by USGS or GAMA databases where the constructed and monitoring well water level measurements met all of the following criteria: (i) the monitoring well and the constructed well were co-located (within one mile (1.61 km) of one another), (ii) the monitoring well and the constructed well were completed at locations with land surface elevations within ±1 m of one another, (iii) the monitoring well and the constructed well were completed to depths within ±1 m of one another, and (iv) the well construction date and the monitoring well water level measurement were separated by no more than 30 days.

Constructed well water level measurements made by well construction companies are, in many (but not all) cases, similar to nearby monitoring well water level measurements by trained technicians or pressure transducers made at similar times, in wells of similar depths (Supplementary Note 3 and Supplementary Figs. 4 and 5). Our comparison implies that water levels reported in well completion reports capture well water level information that is similar to monitoring well water level information. We acknowledge well water elevation data from both monitoring wells and constructed wells have substantial limitations (see ref. [62]).

**Geospatial analyses.** We compared the elevation of water surfaces measured in wells to mean sea level. Uncertainties in our data derive from the land surface elevation dataset[63] (~10 m gridded digital elevation model) and from uncertainties in the locations of groundwater wells. We then calculated the percentage of wells within 10 km of a coast with post-2000 well water elevations that lie below sea level, and report these results in Fig. 4. We completed a suite of sensitivity analyses to demonstrate limits to our analyses arising from (i) the threshold distance from the coast considered in our analysis (Supplementary Note 4), (ii) the presence of non-zero vertical hydraulic gradients and concomitant impacts of well depth on well water elevations (Supplementary Note 4), and (iii) imperfect well location information and resulting increases in well water elevation uncertainties (Supplementary Note 6; for further discussion of limitations arising due to uncertainties in well locations see refs. [61,64]).

We emphasize that our dataset is biased toward places where groundwater wells have been drilled to provide fresh water, because most of our data originate from well completion reports[61]. Thus, the coastal segments that we analyze (i.e., those with at least $n = 10$ well water elevation measurements) capture parts of the coastline where groundwater use is common and where records of groundwater well construction are maintained. Several states do not have publicly available well construction databases, including Georgia, Connecticut, and Rhode Island[61]. Further, imprecise latitude–longitude data in California[64] also hinder our analyses; California wellhead location uncertainty translates to uncertain groundwater level elevations (Supplementary Note 6). Establishing more complete groundwater monitoring and providing more precise well latitude and longitude data (e.g., in California) would enable better assessments of seawater intrusion in these areas.

## Data availability
Well water level datasets are available from state and sub-state agencies. Some states require consent to share groundwater-well data, either through requests to their various agencies, or through public records requests. Supplementary Note 1 includes websites for direct download and contact information for requesting access to the original well completion report data (see also ref. [61]). Monitoring well water level data are available from the US Geological Survey (waterdata.usgs.gov/nwis/inventory) and California's GAMA Program (gamagroundwater.waterboards.ca.gov/gama/gamamap/public).

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

## Author contributions

S.J., D.P., H.S., Y.F., and J.W.K. devised methods, discussed results, and contributed to writing the manuscript; S.J. and D.P. compiled well completion records; S.J., D.P., and H.S. completed geospatial analyses.

## Competing interests

The authors declare no competing interests.
