## [Peer Review File · Nature Communications]

Reviewers' comments:

Reviewer #1 (Remarks to the Author):

In their manuscript, "Groundwater level observations in 250,000 coastal USA wells reveal scope of potential seawater intrusion," Jasechko et al. perform a massive analysis of water levels in wells near the coast and show that groundwater levels are below mean sea level in more than half of all wells along approximately 15% of the contiguous US coastline. The Atlantic and Gulf coasts are more "impacted" than the Pacific coast, likely because the Pacific coast has more relief and therefore more available drawdown before the water level drops below sea level. This analysis appears to be fairly robust, regardless of the range of well depths or distance from the coast used in the analysis, although there are lower water levels closer to the coast and in deeper wells. The extent of the problem was surprising to me, and therein lies the value of the manuscript.

I have two major suggestions:

The authors show in Figures S4 and S5 that the water level observations from drillers' logs are rather coarse (the discrepancies with more accurate USGS and GAMA water levels are often on the order of a meter or so). The density of observations helps compensate for their relatively low accuracy. In contrast, the USGS and GAMA datasets are presumably much more accurate but far less dense. The methods of comparing the datasets were not as direct as I would have hoped, perhaps due to the low density of the USGS and GAMA datasets. I would encourage the authors to consider presenting some calculations, maps, or discussion of the number of USGS and GAMA wells that lie within 10 km of the coast, the number of 20-km coastal segments that they lie in, and the percentage (of the total) that have water levels below sea level. By comparison, how many other wells (from drilling logs) lie in the same 20-km coastal segments populated by a USGS or GAMA well, and what percentage of those wells have water levels below sea level? This would give readers a more direct understanding of why there is a need to use the less accurate drilling records to increase the volume of observations and how the results would change with the smaller, more accurate dataset.

In Figure 2, the distance of 100 km from the coast is such a great length as to be potentially irrelevant to aquifer salinization unless saltwater first travels inland along surface water bodies (as is the case in parts of Florida). As a calculation, assuming a head difference of 10 m (one of the more extreme differences in Figure 2), a reasonably large hydraulic conductivity of 1 cm/s, and porosity of 0.3, it would take 950 years for salt to travel 100 km from the coast to a well. Upcoming may provide a much shorter path to salinization, but it's unclear where the freshwater-saltwater interface lies at a distance of 100 km inland or whether that deeper salt water is even sourced from modern seawater (as opposed to having recharged during previous glacial-interglacial cycles, etc). The authors could address this point by changing the distance from the coast that they analyze in Figure 2 (or any other places where 100 km is the distance of choice—I recognize that smaller distances are used throughout most of the manuscript, including key areas such as Figure 4 and Table 1). Regardless of distance used, it might be useful for the authors to provide some sort of salt travel timescale calculation like the one above to help frame the discussion of salinization more clearly.

A few recommendations are below:

L 121: Please provide range of years wells were drilled (2000-present) in this sentence.

L 126: Please state that the dataset is NED here.

L 129: If the water levels are lower than sea level, there is at least a local landward hydraulic gradient, but not necessarily a widespread regional one. Depending on recharge rate and hydraulic conductivity, there may still be a groundwater divide and seaward hydraulic gradient at the coast that together form a barrier to salinization. This point is alluded to in L 289-293 of the discussion, but it warrants

earlier explanation.

L 133: I would tend to call this concept "equivalent freshwater head" rather than the Ghyben-Herzberg principle but defer to the authors on terminology choice. Suggest quantifying the density effect on equivalent freshwater head. For a 30-m well filled with fresh water, the hydraulic gradient would still be landward unless the water level were greater than 0.6 m above sea level.

Figure 2: Please add scale bars to the panels.

Section S1: Why include inland states in data acquisition information?

In reading the supplemental material, I applaud the authors on their thorough job bringing together large, publicly available datasets that are curated in different ways and have their own unique challenges. I can only imagine what a huge effort this was, and it has led to a very nice result that will be of broad interest to hydrogeologists, decision-makers, and coastal residents.

Review by Audrey Sawyer

Reviewer #2 (Remarks to the Author):

This paper compiles and analyzes millions of groundwater level observations made since 2000 in coastal wells in USA. Through the simple analysis of the data, the authors found that the majority of observed groundwater levels are below sea level along more than 15% of the coastline in the US and concluded that landward hydraulic gradients characterize a substantial fraction of the US coastline. I applaud the authors on their efforts and time to collect such huge groundwater level data. These data, without doubt, indicate potentially that aquifers in some US coasts are under threats of seawater intrusion, and water resources managers should pay attention to the phenomena.

Technically, the most concern I have is that the relationships among seawater intrusion, landward hydraulic gradients, and well groundwater level observations described in the paper. The observations were made within 10 km of the coast. Theoretically, the groundwater level below the sea level in an observation well located several kilometers from the coast does not necessarily cause a landward hydraulic gradient from the coastline, because there could be a hydraulic mound developed between observation wells and the sea due to factors such as topography, artificial recharge, and surface water bodies. Also, previous theoretical studies have demonstrated that if a well pumps groundwater with a safe rate, seawater would not intrude into the well (Strack, 1976). However, even the well pumps groundwater with a safe rate, the groundwater level in a neighboring observation well could be below the sea level because of the drawdown caused by the pumping well.

In Supplementary Section S7, the authors classified the coastal aquifers as "submarine groundwater discharge" and "seawater intrusion or landward gradient". It gives me the feeling that "Seawater intrusion" is equivalent to "landward gradient", and "Seawater intrusion" and "submarine groundwater discharge" are exclusive. First, what is "seawater intrusion" defined here? Steady state or transient? The scales of seawater intrusion in different sites are different. How do you treat this issue? Do you define "seawater intrusion" for conditions under which only the interface moves inland? Second, note that seawater intrusion does not mean that a landward gradient occurs, which has been explained above. Third, under many conditions, seawater intrusion and submarine groundwater discharge are not exclusive and may occur simultaneously in the coastal subsurface. Figure S16 could give one an impression that the analysis based on the groundwater level data well matches largely the results of local-scale studies. However, I think the comparison is somewhat farfetched.

Figure 3a shows that 44% of well water table observations made within 1km of the coast are below sea level. From my point of view, only the wells very close to the coast could show if there is a

landward hydraulic gradient. The authors also need give the number of observations within 1 km of the coast, because they are crucial to conclude if there is a landward hydraulic gradient (or active seawater intrusion).

The authors collected the groundwater level data since the year 2000, but did not show the trend of their variation in the recent 20 years, which I believe is important to water resources managers.

In conclusion, the huge groundwater level data collected are valuable, which could potentially show the risk of extensive seawater intrusion and give a warning to water resources planners and managers in relevant coastal regions. However, using the water level observations made within 10 km of the coast to conjecture the gradient from the coast (or the seawater intrusion condition) is not very rigorous.

Reference

Strack, ODL (1976), Single-potential solution for regional interface problems in coastal aquifers, *Water Resources Research*, 12, 1165-1174.

Reviewer #3 (Remarks to the Author):

I'm attaching this review as a Word file.

Review of Jasechko et al. (2020), " Groundwater level observations in 250,000 coastal USA wells reveal scope of potential seawater intrusion"

Review by A. Fisher, 3/19/20

This timely study presents results from a national (US) analysis of groundwater levels in coastal basins, to estimate the extent of seawater intrusion that may be impacting associated aquifers and coastal communities and aquatic systems. This paper should be published. Please don't misinterpret my detailed questions and suggestions to indicate otherwise. I am interested in this topic, as are many others involved in fresh water research, engineering, and management.

I have some suggestions for clarifying the analyses and language that I hope might provide more context and nuance. Some of these issues are addressed either in the main paper text or the supplement, but I'm suggesting that they be highlighted earlier in the study (in abbreviated form, can be explored in the supplement) to clarify both methods and interpretations of results. I consider these to be minor-to-modest revisions, not major, may not even need rereview.

I am returning a marked-up PDF of the main paper, marking some of these issues and a few smaller suggestions. I am glad to be identified to the authors.

Main concerns in presentation (readily addressable, not a reason to reject!):

(1) As noted, there does not need to be a landward gradient in groundwater level for there to be seawater in an aquifer. But this is not made clear early in the paper. There are two distinct issues that are not commonly understood and should be noted earlier in the text.

(a) Because of the difference in density, even under static conditions there is expected to be a seawater wedge that extends into coastal aquifers at depth (approximated crudely with the idealized

Ghyben-Herzberg relation). Most coastal aquifers were not in static balance before people began to develop GW resources, there was flow to the ocean, and at dynamic steady state, this flow helped to shift the SW wedge seaward and/or change its shape. Still, the pre-development, initial condition in most basins should be a salt wedge of some kind at depth, despite the mean gradient being consistent with flow from land to the ocean. If you add tidal variations, this leads to mixing of FW and SW in the aquifer, as does convection at the salt front. So, coastal aquifers typically have some salty zones near the ocean. It is not clear if one considers this to be "intrusion," per se (definition in paper suggests that intrusion is active movement...but again, that can occur even dynamic steady state in terms of water levels), but it increases vulnerability because that salt is poised for landward advancement. (b) Even with groundwater levels above sea level, a lowering of groundwater level causes disproportionate landward movement of the salty wedge (=intrusion). Simple Ghyben-Herzberg analysis indicates 40:1 ratio of drop in water level to rise in the front of the wedge. Because most aquifers are more horizontal than vertical in dimension, a small rise in wedge interface usually results in a much larger lateral (landward) advance.

For some reason, point (b) continues to elude a lot of engineers, scientists, and resource managers. I was at a public meeting about 10 years ago when a prominent engineer (who had received multiple awards for technical expertise), working as a consultant for an agency, asserted that there was no worry about intrusion in a coastal basin because water levels were above sea level. I could not believe he said this, and when I asked him about it (noting the points above), he dug in on it, would not acknowledge basic physics.

This point is noted briefly in the text (line 62-63, and discussed later more extensively), but I think it needs to be clarified – I suggested a few simple edits in the manuscript text. Note that because of these issues, the current analysis is **conservative**. It highlights the most extreme cases, where there is GW level lower than SL. Intrusion could be more common.

(2) Within coastal basins, active or accelerated seawater intrusion is a consequence of a mass imbalance. At least within CA, this imbalance is a consequence of three main factors: increased GW use (pumpage), shifting land use, changing climate. The first is emphasized in this paper, and certainly this is a common issue. But it is not quite right to talk about intrusion being "caused" by excess pumping. It is pumping in combination with the aggregate of inputs and outputs.

Land use change towards urbanization tends to increase runoff and thus reduce infiltration and recharge (not a hard rule, but a tendency). So, you can hold pumping constant (or even reduce it) but have less recharge and that can lead to intrusion. Ag development in coastal basins can also lead to more runoff (although in the Central Valley, with water imports, there can be more recharge because of deep irrigation returns). Climate change has already resulted in an increase in rainfall intensity in CA (e.g., Russo et al., 2013 and other studies) – this is from the observational record from the last century, not just a simulation prediction. There can be same annual precip, but if more falls in smaller number of more intense storms, we get more runoff, less infiltration, less recharge. Once again, without having to change pumping, we can have GW budget imbalance. Whether the main issue in any particular basin (or part of a basin) is pumping or loss of recharge (from LU or CC) is spatially and temporarily variable – this needs to be assessed basin by basin.

Russo, T. A., A. T. Fisher, and D. W. Winslow (2013), Regional and local increases in storm intensity in the San Francisco Bay Area, USA, between 1890 and 2010, *J. Geophys. Res.-Atm.*, 18, 1-10, doi:10.1002/jgrd.50225.

The following comments are listed in order of appearance

Line 39. "... due to insufficient groundwater level measurements."

This assertion is odd because the next sentence notes that the authors present results from analyzing 250,000 water level measurements. I think that the lack of an earlier assessment is addressed by this study - certainly more data would be helpful, but if there were not enough data, then this study would not have been done. So, the issue really is that no one had previously done all the work required to assemble the data and analyze the results, not a lack of data, per se.

Lines 58-61, especially "...here we focus on the latter."

See main points above. I think these lines are not quite right. The observational data and main analyses are a series of synoptic views of GW levels in coastal basins. Full stop. The link from there to changes in pumpage, climate, and land use are not really addressed in this study, and probably this would require basin/sub-basin scale assessment, and detailed analysis of pumping patterns (in time and space), climate, land use, etc.

Lines 70-110. I would present much of this information after the main analyses are presented, and instead, bring up front a more nuanced explanation of SW intrusion, heterogeneity, and fresh-water heads (e.g., Lines 280-281, 295-301, among others).

Line 98: This line makes a case for the three-dimensional nature of intrusion, but the analysis is really 1D or 2D, especially because of heterogeneity in properties and the distribution of pumping. The issue of heterogeneity is highlighted in Supplement p. 55, for Monterey Bay Region, but I don't think we know how common spatial variability is in intrusion patterns more broadly - my guess is that it is common, because aquifer properties, recharge, and pumping are heterogeneously distributed.

In my region, where SW intrusion is common, the same aquifer system that is intruded in one location is not intruded just a few km away. I've seen this in GW quality records, big differences in chemistry at the same time - one area intruded, nearby area looks fine. Of course, at the large scale, the aquifer is intruded (no question), but I have been faced with upset well owners who argue that the intrusion is not impacting their wells, so they should not be held accountable (financially) for intrusion elsewhere. They have a point. In one case, it appears that un-intruded areas may have wells screened within a paleo channel that is directly connected to a recharge area.

None of this contradicts the key points being made in this paper, but I think some of this nuance is important because aquifers are not continuous, flat layers with uniform properties - Graham Fogg likes to say, "Most of the aquifer is not an aquifer," referring to less conductive regions even within highly transmissive aquifers.

Lines 111-125. It is not clear how biases in where and when data was collected from wells may impact this study. For coastal basins that are worried about SW intrusion, they often emphasize monitoring of wells close to the coast. On the one hand, this could lead to a bias in elevated TDS levels, but water levels tend to be "pinned" by sea level very close to the ocean, so somewhat more inland wells might be more indicative of the lowering of GW levels over time. Also, those coastal networks are likely monitored more frequently than are inland wells. And even if a production well is shut down for measurement, then one nearby may not be.

Lines 193-194, text says, "More than half of all recorded well water levels that are below sea level are found within 8 km of the coast." Is this more than half of well readings or more than half of wells from which readings have been taken?" I think it might be the latter, as Line 199 says, "We report our results as percentages of wells exhibiting landward gradients..." But if this is the case, then language throughout should be checked and clarified. If it is more than half the readings, then there could be a bias if coastal wells are checked more often.

Other examples that are confusing include Lines 215-224 – this section seems to emphasize percentages of water well level observations, rather than wells, from which observations have been made. This pops up again on 227 and 238. Table 1 suggests that these might be percentages of wells, not level measurements. Will be good to clarify.

Not sure if this is possible, but perhaps for selected basins or regions, authors could indicate what fraction of aquifer area is impacted. Maybe that will reduce bias from clustering of wells (in space) and/or inconsistent time of measurements/sampling.

Lines 126-138. As noted in S5, many well records in CA do not include well locations beyond TRS designation (putting wells in the center of a designated Section). This can lead to big errors in estimates of well or screen depth when comparing well "locations" to a highly variable DEM. I'm not sure if/how how this applies in other regions – are well locations in other regions properly georeferenced?

130-134. These are equivalent freshwater gradients, based on equivalent fresh water heads, need to clarify. Point is mentioned on Lines 302-309

Figure 1 caption – seems like too much discussion, better to simplify the caption and refer readers seeking more information to text/supplement sections.

280-281, 295-301 Need to emphasize these considerations earlier in the paper. In many ways, this analysis is conservative in terms of identifying problems.

Line 364. "Abstracting" is an odd usage. How about "extracting" instead?

- What I find especially surprising is that only 15% of the coastline seems to be impacted, particularly when you consider that wells are most common where there is the most development, and monitoring is likely to be most common where there are problems. I would have guessed that 30 or 40 or 50% of coastal basins have problems.

- A final comment on the overall results. It seems like the basis for problems may be different in the three main regions: west, south/southeast, northeast, with coastal aquifers from the latter two areas being more impacted.

- The west is notable for small, sedimentary coastal basins and steep topography and ongoing uplift, with a narrow continental shelf and near-shore oceanic depths. This means that aquifers are exposed to the ocean to considerable depth close to land, but aquifers are highly layered, and shallow aquifers tend to be distinct from deeper ones.

- The south/southeast is notable for wide coastal plains, with a broad continental shelf, could be uplifting or (more likely) sinking due to glacial rebound (uplift to north, sinking to south).

- The northeast is like the south/southeast, but the aquifers there were also subject to recharge at the base of continental ice sheets during the LGM, so those aquifers were somewhat "charged" with fresh water under the shelf (e.g., Person et al., 2003). Now these aquifers are not receiving as much recharge from inland as they were when there were ice sheets.

Person, M., B. Dugan, J. B. Swenson, L. Urbano, C. Stott, J. Taylor, and M. Willett (2003), Pleistocene hydrogeology of the Atlantic continental shelf, New England, *Geol. Soc. Am. Bull.*, 115, 1324–1343.

I wonder how much of the difference between current conditions (more of the south/southeast and northeast indicating conditions favoring intrusion) results from these physiographic and boundary

conditions. Of course, it is more complicated than a single set of conditions: development, climate and other factors must also play roles. But all that said, a broad, thick, flat aquifer seems like it would be more susceptible to intrusion, all else equal...

-
- Reviewer comments are reproduced in **bold black Calibri font**;
 - Author replies are in **blue Calibri font**;
 - Quotes from the authors' revised manuscript are in *purple italicized Calibri font*
-

Reviewers' comments:

Reviewer #1 (Remarks to the Author):

In their manuscript, "Groundwater level observations in 250,000 coastal USA wells reveal scope of potential seawater intrusion," Jasechko et al. perform a massive analysis of water levels in wells near the coast and show that groundwater levels are below mean sea level in more than half of all wells along approximately 15% of the contiguous US coastline. The Atlantic and Gulf coasts are more "impacted" than the Pacific coast, likely because the Pacific coast has more relief and therefore more available drawdown before the water level drops below sea level. This analysis appears to be fairly robust, regardless of the range of well depths or distance from the coast used in the analysis, although there are lower water levels closer to the coast and in deeper wells. The extent of the problem was surprising to me, and therein lies the value of the manuscript.

We thank Dr. Sawyer for her helpful comments. Below we detail modifications and additions made to our manuscript in response to each comment. Our consideration of Dr. Sawyer's comments led us to change our manuscript in ways that, we feel, improved its quality.

I have two major suggestions:

The authors show in Figures S4 and S5 that the water level observations from drillers' logs are rather coarse (the discrepancies with more accurate USGS and GAMA water levels are often on the order of a meter or so). The density of observations helps compensate for their relatively low accuracy. In contrast, the USGS and GAMA datasets are presumably much more accurate but far less dense. The methods of comparing the datasets were not as direct as I would have hoped, perhaps due to the low density of the USGS and GAMA datasets. I would encourage the authors to consider presenting some calculations, maps, or discussion of the number of USGS and GAMA wells that lie within 10 km of the coast, the number of 20-km coastal segments that they lie in, and the percentage (of the total) that have water levels below sea level. By comparison, how many other wells (from drilling logs) lie in the same 20-km coastal segments populated by a USGS or GAMA well, and what percentage of those wells have water levels below sea level? This would give readers a more direct understanding of why there is a need to use the less accurate drilling records to increase the volume of observations and how the results would change with the smaller, more accurate dataset.

We found this suggestion especially helpful because it prompted us to quantify how driller-report water level data compares to USGS/GAMA data both in terms of (a) the density of its spatial distribution, and (b) the frequency with which water level measurements for each dataset lie below sea level. We have added a new Supplementary Section (Section S5); the section compares USGS and GAMA monitoring wells to our driller-report water level data. In Section S5 we present (i) the number of wells within 10 km of the coast for each dataset, following Dr. Sawyer's recommendation to present "the number of USGS and GAMA wells that lie within 10 km of the coast"; (ii) the number of 20 km segments with sufficient

data for analyses in each dataset, following the recommendation to tabulate “the number of 20-km coastal segments that they lie in”; and (iii) the percentage of wells with water levels that lie below sea level using the two datasets separately, following the recommendation to present “the percentage (of the total) that have water levels below sea level”.

(i) We present the number of wells within 10 km of the coast for each dataset, following Dr. Sawyer’s recommendation to present “the number of USGS and GAMA wells that lie within 10 km of the coast”, in tabular form. The table demonstrates the low spatial density of existing monitoring well data, and how the driller-report water levels increase the density of well water level measurements by a factor of 6 to 200 times (West Coast vs Gulf Coast) more than the USGS network. We add the following new supplementary text to complement the new supplementary table presenting these results:

- *“First, we tabulate the number of unique wells reporting at least one water level measurement after the year 2000 among the different datasets. We show that the USGS (and GAMA) dataset—while providing important and locally relevant insights where these wells do exist—is far too sparse to provide useful insights along the vast majority of the U.S. coastline. Specifically, for example, the analyzed USGS data for the entire Gulf Coast has only 706 monitoring wells reporting at least one water level within 10 km of the coast; by contrast, the compiled well completion report water level dataset contains 141,426 post-2000 measurements each in unique wells along the Gulf Coast—200 times the density of the USGS’ network. The driller report water level dataset is 20 times denser than the USGS’ network along the East Coast, and six times denser than the combine USGS/GAMA network along the West Coast (calculated as the number of unique wells with at least one post-2000 water level measurement that exist within 10 km of the coast; see the first row reporting values in Table S3). Thus, including the drillers report dataset greatly increased the data density and the statistical significance of our analyses.”*

Table S3. Wells with post-2000 water level data close to the coast in monitoring well versus well completion report datasets

Maximum distance **	Number of unique wells with at least one post-2000 water level measurement		
	West Coast	Gulf Coast	East Coast
10 km *	USGS/GAMA: 2,059 Driller report: 12,137	USGS/GAMA: 706 Driller report: 141,426	USGS/GAMA: 4,884 Driller report: 95,528
5 km	USGS/GAMA: 1,260 Driller report: 8,057	Driller report: 99,054 USGS/GAMA: 425	Driller report: 59,421 USGS/GAMA: 3,145
2 km	Driller report: 3,622 USGS/GAMA: 628	Driller report: 53,975 USGS/GAMA: 209	Driller report: 30,375 USGS/GAMA: 1,774
1 km	Driller report: 1,893 USGS/GAMA: 358	Driller report: 30,515 USGS/GAMA: 107	Driller report: 18,127 USGS/GAMA: 1,021

* sum of all values in this row (USGS/GAMA plus driller report) is 256,740 (the basis for the title of our paper: “Groundwater level observations in 250,000 coastal USA wells reveal scope of potential seawater intrusion”)

** maximum distance represents the maximum distance from a well (with a post-2000 water level measurement) to the nearest coast.

(ii) We quantify and tabulate the number of 20 km segments with sufficient data for analyses in each dataset (following the recommendation to tabulate “the number of 20-km coastal segments that they lie in”). We include this information in the new supplementary section (S5) in our revised supplementary information via a new figure (Figure S15) that displays the abundance of 20-km-long segments meeting our criteria for analyses using each of the two datasets (top panel: USGS & GAMA, bottom panel: driller report).

(iii) We present “the percentage (of the total) that have water levels below sea level” in the aforementioned new figure, Figure S15, which is pasted below.

In Figure 2, the distance of 100 km from the coast is such a great length as to be potentially irrelevant to aquifer salinization unless saltwater first travels inland along surface water bodies (as is the case in parts of Florida). As a calculation, assuming a head difference of 10 m (one of the more extreme differences in Figure 2), a reasonably large hydraulic conductivity of 1 cm/s, and porosity of 0.3, it would take 950 years for salt to travel 100 km from the coast to a well. Upconing may provide a much shorter path to salinization, but it's unclear where the freshwater-saltwater interface lies at a distance of 100 km inland or whether that deeper salt water is even sourced from modern seawater (as opposed to having recharged during previous glacial-interglacial cycles, etc). The authors could address this point by changing the distance from the coast that they analyze in Figure 2 (or any other places where 100 km is the distance of choice—I recognize that smaller distances are used throughout most of the manuscript, including key areas such as Figure 4 and Table 1).

We thank Dr. Sawyer for her suggestion, and agree with her: portions of aquifers located 100 km from the coast are unlikely to be susceptible to salinization via horizontal migration of intruded seawater over any typical management horizon (decades). We revised the figure. We initially attempted to close in the distance to the coast for which we show point data to 20 km; however, by doing so the black line identifying the 20 km inland distance makes it challenging to see any of the coastal points (see below):

- We have therefore updated our Fig. 2 to show points within 50 km of the coast;
- We kept the six inset maps at the same scale as our first submission in an effort to provide the ‘zoomed out’ perspective required for one to evaluate differences in the extent of water levels that lie below sea level in West versus East/Gulf Coast settings.
- We kept the x-axis as 0-100 km in Fig. 3 to show that well water levels rarely rest below sea level once one moves inland this far; keeping the scale as is (as in Fig. 2 inset maps) provides the reader with an opportunity to put the relatively high fraction of water levels that lie below sea level close to the coast (<1-20 km) into perspective.

The revised figure 2 (50 km buffer from coast) is shown below:

Regardless of distance used, it might be useful for the authors to provide some sort of salt travel timescale calculation like the one above to help frame the discussion of salinization more clearly.

We have added a sentence to our concluding paragraph to make clear that one can expect long time lags for seawater to move inland in many cases, because of the limitations imposed by hydraulic conductivity. We are unsure about applying continental-scale hydraulic conductivity data more broadly as these data products that span the continent (e.g., Gleeson, T. et al. (2014). *Geophysical Research Letters*, 41, 3891-3898), there has yet to be a data output that provides locally relevant data at the spatial resolution of the well completion data (< 1km x,y resolution). To emphasize Dr. Sawyer's point – as it is an important one – we added the following text to our concluding paragraph:

- *“Even where most or all well water elevations lie below sea level, it may take decades for seawater to move kilometers inland, as the hydraulic conductivity of aquifers limits groundwater flow speeds.”*

A few recommendations are below:

L 121: Please provide range of years wells were drilled (2000-present) in this sentence.

Thank you. We added *“measured more recently than the year 2000 and derive..”* to this sentence

L 126: Please state that the dataset is NED here.

Thank you. We added *“digital elevation data from ned.usgs.gov”*

L 129: If the water levels are lower than sea level, there is at least a local landward hydraulic gradient, but not necessarily a widespread regional one. Depending on recharge rate and hydraulic conductivity, there may still be a groundwater divide and seaward hydraulic gradient at the coast that

together form a barrier to salinization. This point is alluded to in L 289-293 of the discussion, but it warrants earlier explanation.

We agree that this point should be made earlier; thank you for suggesting we do so. We added the following text):

- *“Nonetheless, seaward hydraulic gradients can occur even where most well water elevations lie below sea level, if a hydraulic ridge exists (e.g., if well water elevations lie below sea level 5-10 km inland, but are above sea level within 5 km of the coast). Therefore, we evaluate the sensitivity of our results to the distance from the coast (i.e., measurements within 10 km of the coast versus only those within 1 km of the coast; Supplementary Section S4.1), and we find that our main conclusions remain valid within this range of choices.”*

L 133: I would tend to call this concept “equivalent freshwater head” rather than the Ghyben-Herzberg principle but defer to the authors on terminology choice. Suggest quantifying the density effect on equivalent freshwater head. For a 30-m well filled with fresh water, the hydraulic gradient would still be landward unless the water level were greater than 0.6 m above sea level.

We revised to: *“the equivalent hydraulic head resulting from the “Ghyben-Herzberg” principle”*

Figure 2: Please add scale bars to the panels.

Scale bars (see “10 km”) have been added to each of the panels:

Section S1: Why include inland states in data acquisition information?

A good and important question that we can be more clear about. The answer is that we needed these data to evaluate the quality of the well water level measurements reported in driller reports (i.e., data for the entire US were used to compare nearby monitoring well water levels against driller-reported water levels presented in section S3. We should be clearer about this than we were in our first submission; in an effort to do we have added the following text to Section S1:

- *“Although our study focuses only on coastal states, we compiled well water level data for all states (including those inland) in order to develop the largest sample size possible for our comparison of well water levels reported in well completion report versus those measured in monitoring wells (i.e., a larger sample size for our assessment of the quality of driller report water levels presented in Section S3).”*

In our main text (Methods) we include the following revised text:

- *“To evaluate the quality of water levels reported in well completion reports (see the next Methods subsection and Supplementary Section S3), we also compiled well water elevation data for the contiguous United States, Alaska and Hawaii.”*

In reading the supplemental material, I applaud the authors on their thorough job bringing together large, publicly available datasets that are curated in different ways and have their own unique challenges. I can only imagine what a huge effort this was, and it has led to a very nice result that will be of broad interest to hydrogeologists, decision-makers, and coastal residents.

Thank you.

Review by Audrey Sawyer

Reviewer #2 (Remarks to the Author):

This paper compiles and analyzes millions of groundwater level observations made since 2000 in coastal wells in USA. Through the simple analysis of the data, the authors found that the majority of observed groundwater levels are below sea level along more than 15% of the coastline in the US and concluded that landward hydraulic gradients characterize a substantial fraction of the US coastline. I applaud the authors on their efforts and time to collect such huge groundwater level data. These data, without doubt, indicate potentially that aquifers in some US coasts are under threats of seawater intrusion, and water resources managers should pay attention to the phenomena.

We thank Reviewer 2 for the comments and time spent evaluating our manuscript. In the following paragraphs (in our replies in blue text) we detail the amendments made to our work to incorporate the suggestions Reviewer 2 has contributed.

Technically, the most concern I have is that the relationships among seawater intrusion, landward hydraulic gradients, and well groundwater level observations described in the paper. The observations were made within 10 km of the coast. Theoretically, the groundwater level below the sea level in an observation well located several kilometers from the coast does not necessarily cause a landward hydraulic gradient from the coastline, because there could be a hydraulic mound developed between observation wells and the sea due to factors such as topography, artificial recharge, and surface water bodies. Also, previous theoretical studies have demonstrated that if a well pumps groundwater with a safe rate, seawater would not intrude into the well (Strack, 1976). However, even the well pumps groundwater with a safe rate, the groundwater level in a neighboring observation well could be below the sea level because of the drawdown caused by the pumping well.

Reviewer 2's comment is important; we also note that a similar comment was made by Reviewer 1 (Dr. Sawyer). To incorporate this comment, we have added new text to our introduction to make it clear that the 10 km distance inland is indeed arbitrary, but also that we have evaluated the sensitivity of our results to the cutoff distance (i.e., re-running our analysis for all measurements made <2 km from the coast). We emphasize that 10 km is roughly the maximum distance seawater has propagated among site scale studies we compile (e.g., Salinas Valley: seawater has moved inland ~10 km: <http://ccows.csUMB.edu/wiki/index.php/File:2018HistoricSeawaterIntrusionMap180ftAquifer.pdf>). To address Reviewer 2's comment, we also added new text that reads as follows:

- *“Nonetheless, seaward hydraulic gradients can occur even where most well water elevations lie below sea level, if a hydraulic ridge exists (e.g., if well water elevations lie below sea level 5-10 km inland, but are above sea level within 5 km of the coast). Therefore, we evaluate the sensitivity of our results to the distance from the coast (i.e., measurements within 10 km of the coast versus only those within 2 km of the coast; Supplementary Section S4.1), and we find that our main conclusions remain valid within this range of choices.”*

In Supplementary Section S7, the authors classified the coastal aquifers as “submarine groundwater discharge” and “seawater intrusion or landward gradient”. It gives me the feeling that “Seawater intrusion” is equivalent to “landward gradient”, and “Seawater intrusion” and “submarine groundwater discharge” are exclusive. First, what is “seawater intrusion” defined here? Steady state or transient? The scales of seawater intrusion in different sites are different. How do you treat this issue? Do you define “seawater intrusion” for conditions under which only the interface moves inland?

Thank you for this comment. We can improve the clarity of our writing so that the terms are clarified. Our original submission defined seawater intrusion in its first sentence as “...landward fluxes of seawater into aquifers...” (similar to the definition proposed by Dr. A. Werner and colleagues: “...SI refers to the subsurface movement of seawater..”; quotation from the publication: Werner, A. D. et al. (2013). Seawater intrusion processes, investigation and management: recent advances and future challenges. *Advances in water resources*, 51, 3-26). In response to your comment, we have revised our manuscript to clarify that we focus on the potential for seawater intrusion, as it is aligned with what our well water data allow us to evaluate. The potential for seawater intrusion includes areas where landward hydraulic gradients exist, although we acknowledge and clarify in our manuscript that seawater intrusion may also exist where well water elevations are above sea level. Our manuscript introduction has been revised and reads as follows:

- *“Thus, the existence of a landward hydraulic gradient implies that seawater intrusion will occur if the coastal aquifer is well connected to the sea. Identifying locations with such landward hydraulic gradients can help identify aquifers that are susceptible to seawater intrusion, because hydraulic gradients drive groundwater flow, and because coastal hydraulic gradients influence the depth at which aquifers transition from fresh to brackish water⁶⁻⁸.”*

Second, note that seawater intrusion does not mean that a landward gradient occurs, which has been explained above.

Thank you. We agree and provide the following text in our manuscript to enhance clarification:

- *“Seawater intrusion can occur before landward hydraulic gradients form, because seawater’s higher density can cause it to move landward even when coastal groundwater levels are above sea level.”*

Third, under many conditions, seawater intrusion and submarine groundwater discharge are not exclusive and may occur simultaneously in the coastal subsurface. Figure S16 could give one an impression that the analysis based on the groundwater level data well matches largely the results of local-scale studies. However, I think the comparison is somewhat farfetched.

We thank Reviewer 2 for their comment. Our intention for creating Figure S16 was to highlight the local relevance of the well water level data we compiled in a way that honors the decades of research on seawater intrusion and submarine groundwater discharge that precede our paper. We recognize, based on Reviewer 2’s comment, that we can add additional clarification to make clear that seawater intrusion and submarine groundwater discharges are not mutually exclusive. To do so, we add the following

statement to Supplementary Section S8, which is the section in which the figure (now Fig. S21) identified by Reviewer 2 is located:

- *“We emphasize that seawater intrusion and submarine groundwater discharge are not mutually exclusive at the spatial scale we analyze.”*

We made another modification to our manuscript in consideration of the Reviewer’s important comment; we added text highlighting that the way in which we group sites in space (20 km-long coastline segments) could include portions of the coast characterized by seaward fluxes and other portions of the same coastline segment with landward gradients (i.e., a given 20 km-long coastline segment passes through one area with seaward fluxes, then another farther along the coast characterized by landward fluxes). The new text reads:

- *“Further, the spatial scale at which we complete our analyses (e.g., 10 km inland from the coast and 20 km-long coastline segments) may include places characterized by seaward hydraulic gradients and some other places characterized by landward hydraulic gradients.”*

Figure 3a shows that 44% of well water table observations made within 1km of the coast are below sea level. From my point of view, only the wells very close to the coast could show if there is a landward hydraulic gradient. The authors also need give the number of observations within 1 km of the coast, because they are crucial to conclude if there is a landward hydraulic gradient (or active seawater intrusion).

Thank you for this important comment. Our initial submission provided a sensitivity analysis (i.e., reproduced our results) using distances inland of 10 km, 5 km and 2 km. Based on Reviewer 2’s comment, we added a new sensitivity analysis: results using only those water level measurements made within 1 km of the coast (Supplementary Section S4.1):

Figure S6. Well water level elevations relative to sea level along contiguous U.S. coasts. The map in main text Figure 4 displays fractional distances along each coast corresponding to the y-axes (West and East Coasts) and x-axis (Gulf Coast). Colors along the coastlines represent 20 km segments of coast; orange segments represent those where at least half of the measured well water levels are below sea level and blue segments are those where more than half of well water levels are above sea level. Our analysis was completed using well water level measurements made within 5 km of a coastline (note: results presented in the main text were calculated using well water level measurements made within 10 km of the nearest coast). The three bar plots present the fraction of well water level measurements that are below sea level; each bar represents one 20 km coastline segment. We only display results for coastline segments with at least 10 well water level measurements made within 5 km of the coast.

We found it interesting that in the Salinas Valley, seawater has intruded several miles inland: <http://ccows.csusb.edu/wiki/index.php/File:2018HistoricSeawaterIntrusionMap180ftAquifer.pdf>. In consideration of the literature on this topic, which suggests that there is support for sensitivity analyses that include a range of distances in addition to 1 km, we include the 2 km, 5 km, and 10 km sensitivity analyses, too.

The authors collected the groundwater level data since the year 2000, but did not show the trend of their variation in the recent 20 years, which I believe is important to water resources managers.

We agree that this is important to water managers. Unfortunately, due to page limits, we could not include the trend analysis in the main text. In Fig. S21, we present well water level variations among monitoring wells near the coast to identify areas where water levels have increased versus decreased over time (Fig. S21 reproduced below).

In conclusion, the huge groundwater level data collected are valuable, which could potentially show the risk of extensive seawater intrusion and give a warning to water resources planners and managers in relevant coastal regions.

Thank you.

However, using the water level observations made within 10 km of the coast to conjecture the gradient from the coast (or the seawater intrusion condition) is not very rigorous.

We agree, and to account for this comment we performed extensive sensitivity analyses varying the distance to the coast. To ensure it is clear to readers that we analyze measurements made closer to the coast (i.e., isolating only those measurements within 1 and 2 km of the coast) we include the following new text in our revised manuscript: *“Therefore, we evaluate the sensitivity of our results to the distance from the coast (i.e., measurements within 10 km of the coast versus only those within 2 km of the coast;*

Supplementary Section S4.1), and we find that our main conclusions remain valid within this range of choices.”

There are places where seawater has intruded ~10 km (<http://ccows.csUMB.edu/wiki/index.php/File:2018HistoricSeawaterIntrusionMap180ftAquifer.pdf>). It is important we make additions to the manuscript to include the reviewer’s comment; we have added another threshold (1 km) to our sensitivity analyses to highlight how our results vary as we consider greater distances inland from the coast (table below reproduced from Supplementary Information):

Table S3. Wells with post-2000 water level data close to the coast in monitoring well versus well completion report datasets

Maximum distance **	Number of unique wells with at least one post-2000 water level measurement		
	West Coast	Gulf Coast	East Coast
10 km *	USGS/GAMA: 2,059 Driller report: 12,137	USGS/GAMA: 706 Driller report: 141,426	USGS/GAMA: 4,884 Driller report: 95,528
5 km	USGS/GAMA: 1,260 Driller report: 8,057	Driller report: 99,054 USGS/GAMA: 425	Driller report: 59,421 USGS/GAMA: 3,145
2 km	Driller report: 3,622 USGS/GAMA: 628	Driller report: 53,975 USGS/GAMA: 209	Driller report: 30,375 USGS/GAMA: 1,774
1 km	Driller report: 1,893 USGS/GAMA: 358	Driller report: 30,515 USGS/GAMA: 107	Driller report: 18,127 USGS/GAMA: 1,021

* sum of all values in this row (USGS/GAMA plus driller report) is 256,740 (the basis for the title of our paper: “Groundwater level observations in 250,000 coastal USA wells reveal scope of potential seawater intrusion”)

** maximum distance represents the maximum distance from a well (with a post-2000 water level measurement) to the nearest coast.

Reference

Strack, ODL (1976), Single-potential solution for regional interface problems in coastal aquifers, *Water Resources Research*, 12, 1165-1174.

Reviewer #3 (Remarks to the Author):

I'm attaching this review as a Word file.

Review of Jasechko et al. (2020), " Groundwater level observations in 250,000 coastal USA wells reveal scope of potential seawater intrusion"

Review by A. Fisher, 3/19/20

We thank Dr. Fisher for their suggestions; incorporating these has made our manuscript better.

Below (in blue text) we detail additions and modifications made to the manuscript in response to each of the overarching comments provided by Dr. Fisher. At the end of these official comments, we copy and pasted each of the in-line comments (i.e., those embedded in the *.pdf file that was provided to us by the journal) and describe the adjustments and additions we have made in response to those comments to ensure we do not overlook any comments derived from either of the two attachments provided to us as part of Dr. Fisher's review.

This timely study presents results from a national (US) analysis of groundwater levels in coastal basins, to estimate the extent of seawater intrusion that may be impacting associated aquifers and coastal communities and aquatic systems. This paper should be published. Please don't misinterpret my detailed questions and suggestions to indicate otherwise. I am interested in this topic, as are many others involved in fresh water research, engineering, and management.

I have some suggestions for clarifying the analyses and language that I hope might provide more context and nuance. Some of these issues are addressed either in the main paper text or the supplement, but I'm suggesting that they be highlighted earlier in the study (in abbreviated form, can be explored in the supplement) to clarify both methods and interpretations of results. I consider these to be minor-to-modest revisions, not major, may not even need rereview.

I am returning a marked-up PDF of the main paper, marking some of these issues and a few smaller suggestions. I am glad to be identified to the authors.

Thank you.

Main concerns in presentation (readily addressable, not a reason to reject!):

(1) As noted, there does not need to be a landward gradient in groundwater level for there to be seawater in an aquifer. But this is not made clear early in the paper. There are two distinct issues that are not commonly understood and should be noted earlier in the text.

(a) Because of the difference in density, even under static conditions there is expected to be a seawater wedge that extends into coastal aquifers at depth (approximated crudely with the idealized Ghyben-Herzberg relation). Most coastal aquifers were not in static balance before people began to develop GW resources, there was flow to the ocean, and at dynamic steady state, this flow helped to shift the SW wedge seaward and/or change its shape. Still, the pre-development, initial condition in

most basins should be a salt wedge of some kind at depth, despite the mean gradient being consistent with flow from land to the ocean. If you add tidal variations, this leads to mixing of FW and SW in the aquifer, as does convection at the salt front. So, coastal aquifers typically have some salty zones near the ocean. It is not clear if one considers this to be "intrusion," per se (definition in paper suggests that intrusion is active movement...but again, that can occur even dynamic steady state in terms of water levels), but it increases vulnerability because that salt is poised for landward advancement.

Thank you for each of these two main points. We've made additions to improve how we communicate these important concepts—as we agree some are not widely understood. We have added a new sentence to our second paragraph (as early as we could without disrupting the flow of the introduction); it reads:

- *“Even under static, pre-development conditions one would expect seawater to exist at depth beneath low-lying coastal lands, because seawater has a higher density than freshwater and because tidal variations disperse the fresh-saline interface in coastal aquifers^{1,2}.“*

(b) Even with groundwater levels above sea level, a lowering of groundwater level causes disproportionate landward movement of the salty wedge (=intrusion). Simple Ghyben-Herzberg analysis indicates 40:1 ratio of drop in water level to rise in the front of the wedge. Because most aquifers are more horizontal than vertical in dimension, a small rise in wedge interface usually results in a much larger lateral (landward) advance.

For some reason, point (b) continues to elude a lot of engineers, scientists, and resource managers. I was at a public meeting about 10 years ago when a prominent engineer (who had received multiple awards for technical expertise), working as a consultant for an agency, asserted that there was no worry about intrusion in a coastal basin because water levels were above sea level. I could not believe he said this, and when I asked him about it (noting the points above), he dug in on it, would not acknowledge basic physics.

This is an important point; thank you. We added the following statement to our manuscript:

- *“We emphasize that a small drop in the elevation of a water table—which may still lie above sea level—can lead to substantial landward movements of seawater where topographic relief is minimal.”*

This point is noted briefly in the text (line 62-63, and discussed later more extensively), but I think it needs to be clarified – I suggested a few simple edits in the manuscript text. Note that because of these issues, the current analysis is **conservative**. It highlights the most extreme cases, where there is GW level lower than SL. Intrusion could be more common.

We agree. We provide text early in our manuscript to this effect (*“As a result, our approach—detecting landward hydraulic gradients and interpreting these as precursors to seawater intrusion—is conservative.”*) but feel it is worthwhile re-emphasizing this, especially in light of this comment. We added new text to our final (concluding paragraphs) in an effort to further embed Dr. Fisher’s point into our manuscript and provide a reminder to readers that we may be underestimating the severity of the issue here. The new text reads as follows:

- *“We re-emphasize that seawater intrusion can occur even where most well water elevations lie above sea level, meaning our main finding—that 15% of US coastlines are characterized by landward hydraulic gradients—probably underestimates the prevalence of vulnerability to seawater intrusion.”*

(2) Within coastal basins, active or accelerated seawater intrusion is a consequence of a mass imbalance. At least within CA, this imbalance is a consequence of three main factors: increased GW use (pumpage), shifting land use, changing climate. The first is emphasized in this paper, and certainly this is a common issue. But it is not quite right to talk about intrusion being "caused" by excess pumping. It is pumping in combination with the aggregate of inputs and outputs.

Land use change towards urbanization tends to increase runoff and thus reduce infiltration and recharge (not a hard rule, but a tendency). So, you can hold pumping constant (or even reduce it) but have less recharge and that can lead to intrusion. Ag development in coastal basins can also lead to more runoff (although in the Central Valley, with water imports, there can be more recharge because of deep irrigation returns). Climate change has already resulted in an increase in rainfall intensity in CA (e.g., Russo et al., 2013 and other studies) – this is from the observational record from the last century, not just a simulation prediction. There can be same annual precip, but if more falls in smaller number of more intense storms, we get more runoff, less infiltration, less recharge. Once again, without having to change pumping, we can have GW budget imbalance. Whether the main issue in any particular basin (or part of a basin) is pumping or loss of recharge (from LU or CC) is spatially and temporarily variable – this needs to be assessed basin by basin.

Russo, T. A., A. T. Fisher, and D. W. Winslow (2013), Regional and local increases in storm intensity in the San Francisco Bay Area, USA, between 1890 and 2010, *J. Geophys. Res.-Atm.*, **18**, 1-10, doi:10.1002/jgrd.50225.

Thank you for these comments. We agree that the mass balance is what really matters, and that groundwater withdrawals form one flux but that others are critical, too. We have added a new sentence to our second paragraph of the introduction that reads: *“Reductions in recharge—potentially arising via land use changes or climate variations—can reduce groundwater levels and induce seawater intrusion.”*

The following comments are listed in order of appearance

Line 39. "... due to insufficient groundwater level measurements."

This assertion is odd because the next sentence notes that the authors present results from analyzing 250,000 water level measurements. I think that the lack of an earlier assessment is addressed by this study - certainly more data would be helpful, but if there were not enough data, then this study would not have been done. So, the issue really is that no one had previously done all the work required to assemble the data and analyze the results, not a lack of data, per se.

Thank you – we have deleted the highlighted statement (which previously read as): “, due to insufficient groundwater level measurements”

Lines 58-61, especially "...here we focus on the latter."

See main points above. I think these lines are not quite right. The observational data and main

analyses are a series of synoptic views of GW levels in coastal basins. Full stop. The link from there to changes in pumpage, climate, and land use are not really addressed in this study, and probably this would require basin/sub-basin scale assessment, and detailed analysis of pumping patterns (in time and space), climate, land use, etc.

We agree, and making these changes made our manuscript better. Thank you. The following are copied and pasted from earlier reply to comment for convenience:

1. We have deleted the statement (included in our first submission, and highlighted as a comment by Dr. Fisher in their annotated *.pdf file) “Here we focus on the latter.”
2. We include in our second paragraph of our revised manuscript the following: *“Reductions in recharge—potentially arising via land use changes or climate variations—can reduce groundwater levels and induce seawater intrusion.”*

Lines 70-110. I would present much of this information after the main analyses are presented, and instead, bring up front a more nuanced explanation of SW intrusion, heterogeneity, and fresh-water heads (e.g., Lines 280-281, 295-301, among others).

We considered the placement of the discussion of our local-scale studies at length. We decided that it would be appropriate and respectful to authors of these studies to review these works early in the manuscript; our rationale is that by doing so early on, we can show what’s been done, what we know and thereby set up the knowledge gap that justifies our study. The change we have incorporated to embed Dr. Fisher’s comment into our manuscript is to add the suggested further explanation of seawater intrusion and heterogeneity and freshwater heads into this introductory paragraph. Specifically, we add the following to our introductory paragraph:

- *“A common theme among the local-scale studies we reviewed is that the rates and the vertical and lateral network of intruding seawater depend not only on hydraulic gradients but also, critically, on the heterogeneity and architecture of the coastal aquifer system (i.e., connectivity of sedimentary layers to the sea, presence of “windows” (i.e., gaps) in aquitards, etc.). “*

Line 98: This line makes a case for the three-dimensional nature of intrusion, but the analysis is really 1D or 2D, especially because of heterogeneity in properties and the distribution of pumping. The issue of heterogeneity is highlighted in Supplement p. 55, for Monterey Bay Region, but I don’t think we know how common spatial variability is in intrusion patterns more broadly – my guess is that it is common, because aquifer properties, recharge, and pumping are heterogeneously distributed.

We agree: rarely, if ever, do scientists underestimate the heterogeneity of aquifer systems. We hope that by beginning our manuscript with a discussion of some of these important local scale studies and providing a set of conceptual models (which themselves are oversimplifications of actual heterogeneities in these systems) we begin to convey some of this to readers less familiar with the extent of variability in the subsurface.

In my region, where SW intrusion is common, the same aquifer system that is intruded in one location is not intruded just a few km away. I've seen this in GW quality records, big differences in chemistry at the same time – one area intruded, nearby area looks fine. Of course, at the large scale, the aquifer is intruded (no question), but I have been faced with upset well owners who argue that the intrusion is

not impacting their wells, so they should not be held accountable (financially) for intrusion elsewhere. They have a point. In one case, it appears that un-intruded areas may have wells screened within a paleo channel that is directly connected to a recharge area.

None of this contradicts the key points being made in this paper, but I think some of this nuance is important because aquifers are not continuous, flat layers with uniform properties – Graham Fogg likes to say, "Most of the aquifer is not an aquifer," referring to less conductive regions even within highly transmissive aquifers.

These are helpful points. Thank you.

We added new text to try to incorporate this point while also not disrupting the flow of this paragraph (which is intended to focus on the vertical dimension):

- *"...just as they may differ laterally at different locations situated along the coastline".*

Because we felt we could not make this point as clearly as we would like without disrupting the rhythm of the current paragraph, we added some more text elsewhere; specifically, we added this new statement to the discussion:

- *"Further, heterogeneity in aquifer flow paths and connectivity to surface processes mean that hydraulic heads may switch from below to above sea level over short lateral distances along coastlines (~kilometers); thus, even in sections of coastline where the great majority of well water elevations lie below sea level, not all wells are necessarily vulnerable to seawater intrusion."*

Lines 111-125. It is not clear how biases in where and when data was collected from wells may impact this study. For coastal basins that are worried about SW intrusion, they often emphasize monitoring of wells close to the coast. On the one hand, this could lead to a bias in elevated TDS levels, but water levels tend to be "pinned" by sea level very close to the ocean, so somewhat more inland wells might be more indicative of the lowering of GW levels over time. Also, those coastal networks are likely monitored more frequently than are inland wells. And even if a production well is shut down for measurement, then one nearby may not be.

This is an interesting point. In some ways, the results of our sensitivity analyses (i.e., varying the distance inland) may capture some of this in that the more expansive distances may allow for greater numbers of inland wells that are 'unpinned'.

Lines 193-194, text says, "More than half of all recorded well water levels that are below sea level are found within 8 km of the coast." Is this more than half of well readings or more than half of wells from which readings have been taken?" I think it might be the latter, as Line 199 says, "We report our results as percentages of wells exhibiting landward gradients..." But if this is the case, then language throughout should be checked and clarified. If it is more than half the readings, then there could be a bias if coastal wells are checked more often.

Other examples that are confusing include Lines 215-224 – this section seems to emphasize percentages of water well level observations, rather than wells, from which observations have been

made. This pops up again on 227 and 238. Table 1 suggests that these might be percentages of wells, not level measurements. Will be good to clarify.

Your intuition is correct (each well is counted but only once); and, we can and should be more clear. To remedy we have added:

1. New text to the caption of Figure 3 that reads: *“Each well is counted only once; for wells with multiple measurements postdating 2000, we calculated median well water elevations for 2000-present and incorporate only this median value in our calculations to produce the above figures.”*

2. We add the following sentence (now sentence #2 of our results, so that this is clear right from the get go): *“For monitoring wells reporting more than one water elevation measurement, we include only the median of all measurements made from 2000-present in all results to follow.”*

Not sure if this is possible, but perhaps for selected basins or regions, authors could indicate what fraction of aquifer area is impacted. Maybe that will reduce bias from clustering of wells (in space) and/or inconsistent time of measurements/sampling.

We were careful to ‘bin’ coastal areas into set distance intervals when reporting our statistics; this ensured areas with more well water level measurements were not unfairly weighted in our results, as would be the case if we focused our discussion purely on the number of well water levels that lie below sea level). By reporting the fraction of well water levels that lie below sea level in our Fig. 4, we attempt to make Dr. Fisher’s point to readers: that these systems are not binary; i.e., that a portion of well water levels may lie below sea level while others are above sea level, even in areas where seawater intrusion has been identified. We’ve added a new sentence to highlight how our spatial sampling could include sections of coastline characterized by landward gradients and also another area(s) characterized by a seaward gradient:

- *“Further, the spatial scale at which we complete our analyses (e.g., 10 km inland from the coast and 20 km-long coastline segments) may include places characterized by seaward hydraulic gradients and some other places characterized by landward hydraulic gradients.”*

Lines 126-138. As noted in S5, many well records in CA do not include well locations beyond TRS designation (putting wells in the center of a designated Section). This can lead to big errors in estimates of well or screen depth when comparing well "locations" to a highly variable DEM. I'm not sure if/how how this applies in other regions – are well locations in other regions properly georeferenced?

For the most part: yes, well locations from drilling reports are GPS referenced. This is especially true as we restrict our analysis to measurements made since the year 2000. There are a few exceptions (e.g. Broward County) that we are careful to document and make transparent in the supplementary information.

130-134. These are equivalent freshwater gradients, based on equivalent fresh water heads, need to clarify. Point is mentioned on Lines 302-309

Thank you. Dr. Sawyer makes a similar recommendation to include the term equivalent freshwater head or similar here. We do so in our revised manuscript, adding *“the equivalent hydraulic head”*.

Figure 1 caption – seems like too much discussion, better to simplify the caption and refer readers seeking more information to text/supplement sections.

We have reduced the word count of our Fig. 1 caption from (initial submission) 454 words to (this revised submission) 328 words.

280-281, 295-301 Need to emphasize these considerations earlier in the paper. In many ways, this analysis is conservative in terms of identifying problems.

We agree – thank you. We have (1) added new text to the introduction reading: *“We emphasize that a small drop in the elevation of a water table—which may still lie above sea level—can lead to substantial landward movements of seawater where topographic relief is minimal.”* and (2) re-emphasized how our analysis is conservative (to supplement our statement in the introduction about the conservative nature of our results); the new text reads: *“We re-emphasize that seawater intrusion can occur even where most well water elevations lie above sea level, meaning our main finding—that 15% of US coastlines are characterized by landward hydraulic gradients—probably underestimates the prevalence of vulnerability to seawater intrusion.”*

Line 364. "Abstracting" is an odd usage. How about "extracting" instead?

Thank you; we have replaced with *“Extracting”*

- What I find especially surprising is that only 15% of the coastline seems to be impacted, particularly when you consider that wells are most common where there is the most development, and monitoring is likely to be most common where there are problems. I would have guessed that 30 or 40 or 50% of coastal basins have problems.

We too found these results intriguing; Dr. Sawyer makes a good point in their review, that the *“..Pacific coast has more relief and therefore more available drawdown before the water level drops below sea level”*. We agree with Dr. Fisher—many low-elevation basins along the west coast appear to have landward hydraulic gradients.

- A final comment on the overall results. It seems like the basis for problems may be different in the three main regions: west, south/southeast, northeast, with coastal aquifers from the latter two areas being more impacted.

- The west is notable for small, sedimentary coastal basins and steep topography and ongoing uplift, with a narrow continental shelf and near-shore oceanic depths. This means that aquifers are exposed to the ocean to considerable depth close to land, but aquifers are highly layered, and shallow aquifers tend to be distinct from deeper ones.

- The south/southeast is notable for wide coastal plains, with a broad continental shelf, could be uplifting or (more likely) sinking due to glacial rebound (uplift to north, sinking to south).

- The northeast is like the south/southeast, but the aquifers there were also subject to recharge at the base of continental ice sheets during the LGM, so those aquifers were somewhat "charged" with fresh water under the shelf (e.g., Person et al., 2003). Now these aquifers are not receiving as much recharge from inland as they were when there were ice sheets.

Person, M., B. Dugan, J. B. Swenson, L. Urbano, C. Stott, J. Taylor, and M. Willett (2003), Pleistocene hydrogeology of the Atlantic continental shelf, New England, Geol. Soc. Am. Bull., 115, 1324–1343.

I wonder how much of the difference between current conditions (more of the south/southeast and northeast indicating conditions favoring intrusion) results from these physiographic and boundary conditions. Of course, it is more complicated than a single set of conditions: development, climate and other factors must also play roles. But all that said, a broad, thick, flat aquifer seems like it would be more susceptible to intrusion, all else equal...

We agree with their conceptual model. We hope that some of this may come out in reader's interpretations of Fig. 1 (i.e., conceptual models juxtaposing high-relief settings along the West Coast versus those low-elevation coastal plain aquifers along the US East and Gulf Coasts).

Incorporating your comments improved our manuscript; thank you.

--- --- --- ---

*the following text derive from the *.pdf provided to us. Each **statement in bold text** is a transcription of a comment provided by Dr. Fisher (we have also provided the line number where each comment was originally made). We only re-post comments that were not included in the main review, and do so to ensure we adequately address each suggestion. The *blue text* beneath each transcribed comments details the change we have made to our manuscript in response to the comment, and any quotes from our revised manuscript are displayed in *purple text*.*

Line 37: addition suggested: “ , but intrusion can occur due to a lowering of groundwater level near the coast, even if the groundwater level remains above sea level.”

We have added text to our manuscript's summary paragraph. We ended up rewriting the entire sentence in an effort to include the suggested text while also keeping our sentences in this summary paragraph as efficient as possible. Our revised sentence reads: *“Seawater intrusion is particularly likely where water tables lie below sea level, but can also arise from groundwater pumping in some coastal aquifers with water tables above sea level”*

Line 54: suggested addition: “contaminates a much larger volume of groundwater and”

We have added the statement as suggested: *“contaminates a much larger volume of groundwater and”*

Line 59: comment “Not really” (highlighting text in our original manuscript that stated “Here we focus on the latter”).

We have deleted the statement “Here we focus on the latter.”

Line 62: minor wording changes suggested (e.g., strike “even”)

We have made adjustments to this sentence where suggested.

Line 94-95: add a space between '30' and 'm'

We have done so (thank you).

Line 98 comment: "I agree that the processes and impacts are three dimensional, but the authors should also tag lateral heterogeneity. In my region, where intrusion is common, the same aquifer system that is intruded in one location (x,y) is not intruded just a few km away. In fact, intrusion and GW outflow to the ocean are occurring simultaneously from this system. Of course, at the large scale, the aquifer is intruded (no question), but I have been faced with upset well owners who argue that the intrusion is not impacting their wells. They have a point. Their wells may be screened within a paleo channel that is directly connected to a recharge area. None of this contradicts the key points being made, but I think some of this nuance is important."

A fair and good point. Thank you. We added new text to incorporate this point while also not disrupting the flow of this paragraph (which is intended to focus on the vertical dimension): *"...just as they may differ laterally at different locations situated along the coastline (e.g., the distance seawater has intruded inland differs between the "180-foot Aquifer" and the "400-foot Aquifer" in the Salinas Valley of California¹⁵⁻¹⁷; Fig. 1 left panel)."*

Because we felt we could not make this point as clearly as we would like without disrupting the rhythm of the current paragraph, we added text elsewhere; specifically, we added this new statement to the discussion: *"Further, heterogeneity in aquifer flow paths and connectivity to surface processes mean that hydraulic heads may switch from below to above sea level over short lateral distances along coastlines (~kilometers); thus, even in sections of coastline where the great majority of well water elevations lie below sea level, not all wells are necessarily vulnerable to seawater intrusion."*

Line 194 additional text suggested: "in the study region"

Thank you; we have made this addition with a minor revision intended to bolster clarity (as to what the study region really represents). The added text reads: *"in our compiled dataset of US well water levels"*

REVIEWERS' COMMENTS:

Reviewer #1 (Remarks to the Author):

I feel the authors have thoroughly addressed the reviews.
--Audrey Sawyer

Reviewer #2 (Remarks to the Author):

The authors have well addressed my comments (and the comments made by other two reviewers), and I am satisfied with the revision made. I think the paper could be accepted. This paper will draw highly attention of practitioners and researchers to coastal groundwater issues.

Chunhui Lu

Reviewer #3 (Remarks to the Author):

The authors of this study have improved the presentation through revision, and I recommend publishing it in Nature Communications.

I appreciate the authors' care in addressing questions raised by reviewers of the previous submission. Reading responses to other reviewer comments, beyond my own, was especially helpful. I also thank the authors for being so careful in presenting methodology and supporting data and plots in the supplement. There is also a lot of nuance added in the explanations, for example, that there can be SWI and SGD simultaneously, which can be confusing. The revision to language has helped to clarify complex issues that will be of special concern to experts, while highlighting the key results that will be of broad interest to Nature Communications readers.

I have no additional comments and encourage publication of this work.

best wishes, Andy Fisher